# Bridging Jensen Gap for Max-Min Group Fairness Optimization in Recommendation

**Chen Xu[1], Yuxin Li[1], Wenjie Wang[2], Liang Pang[3], Jun Xu[1]\*, Tat-Seng Chua[4]**

[1] Gaoling School of Artificial Intelligence, Renmin University of China, Beijing, China
[2] School of Information Science and Technology,
University of Science and Technology of China, Hefei, China
[3] Institute of Computing Technology, Chinese Academy of Sciences, Beijing, China,
[4] NExT++ Research Center, National University of Singapore, Singapore
{xc_chen,yuxin_li}@ruc.edu.cn, wenjiewang96@gmail.com,
pangliang@ict.ac.cn, junxu@ruc.edu.cn, dcscts@nus.edu.sg

## Abstract

Group max-min fairness (MMF) is commonly used in fairness-aware recommender systems (RS) as an optimization objective, as it aims to protect marginalized item groups and ensures a fair competition platform. However, our theoretical analysis indicates that integrating MMF constraint violates the assumption of sample independence during optimization, causing the loss function to deviate from linear additivity. Such nonlinearity property introduces the Jensen gap between the model's convergence point and the optimal point if mini-batch sampling is applied. Both theoretical and empirical studies show that as the mini-batch size decreases and the group size increases, the Jensen gap will widen accordingly. Some methods using heuristic re-weighting or debiasing strategies have the potential to bridge the Jensen gap. However, they either lack theoretical guarantees or suffer from heavy computational costs. To overcome these limitations, we first theoretically demonstrate that the MMF-constrained objective can be essentially reformulated as a group-weighted optimization objective. Then we present an efficient and effective algorithm named FairDual, which utilizes a dual optimization technique to minimize Jensen gap. Our theoretical analysis demonstrates that FairDual can achieve a sub-linear convergence rate to the globally optimal solution and the Jensen gap can be well bounded under a mini-batch sampling strategy with random shuffle. Extensive experiments conducted using six large-scale RS backbone models on three publicly available datasets demonstrate that FairDual outperforms all baselines in terms of both accuracy and fairness. Our data and codes are shared at https://github.com/XuChen0427/FairDual.

## 1 Introduction

Group max-min fairness (MMF) has gained significant attention in industrial recommender systems (RS), as it seeks to provide support for marginalized item groups, where item groups are usually divided by item categories or providers (Xu et al., 2023; Patro et al., 2020; Xu et al., 2024b). For example, optimizing MMF can alleviate the low exposure problem of small sellers in the Amazon RS platform (Patro et al., 2020). As illustrated in European competition law (Jones et al., 2014), protecting these weak supplier groups is essential for preventing large platforms from engaging in unfair strategies, thereby ensuring fair competition (Matten et al., 2008).

Formally, the group MMF optimization objective entails calculating the overall utility of each group and maximizing the utility of the least advantaged group. Typically, group utility computation often necessitates aggregating the overall recommendation ranking list over a specified period (Xu et al., 2023; Do et al., 2021; Xu et al., 2024b), for example, the group utility could be the exposures of action movies within a day. Due to the limited ranking slots, adjusting the exposure of items to improve the

---

\*Corresponding author. Work partially done at Engineering Research Center of Next-Generation Intelligent Search and Recommendation, Ministry of Education.

utility of one item group will inevitably affect the ranking outcomes of other item groups. Therefore, the loss function with the MMF constraint for RS violates a crucial assumption: the independence of samples, resulting in the MMF loss of different item groups not adhering to linear additivity (see theoretical proof in Section 4).

We theoretically and empirically show that the non-linear additivity property of the MMF-constrained objective will introduce a Jensen gap (Gao et al., 2017; Ullah et al., 2021) between the model's convergence point and the optimal point when if mini-batch sampling is applied for optimization (see Section 4). Meanwhile, it is proved that as the mini-batch size decreases and the group size increases, the Jensen gap will become more pronounced in the optimization process, significantly harming the model's performance. However, mini-batch sampling strategies are essential for accelerating the model training process, especially as data and model sizes in RS continue to grow, such as the development of large language models (LLMs) based RS (Bao et al., 2023; Lin et al., 2024a;b).

Some previous approaches have the potential to bridge the Jensen gap for RS. One type of heuristic approach can be applied to bridge the Jensen gap, such as sample re-weighting strategies (Chen et al., 2023b; Wen et al., 2022), which dynamically assigns a higher weight to the weaker group across different batches. However, the effectiveness of this research line is limited due to the lack of theoretical guarantees. Another type of work utilizes machine learning techniques that can help to optimize the non-linear additive loss functions. For example, Abernethy et al. (2022); Cousins (2022) have proposed sampling strategies to obtain unbiased samples, while some methods utilize debiasing gradient descent (Demidovich et al., 2023; Agarwal et al., 2018) to introduce a bias correction term in the gradient. However, these methods cannot be applied to existing large-scale industrial RS, as they often require a convex optimization process that is impractical for RS that typically serves millions of users and hundreds of groups.

To overcome the challenges for bridging the Jensen gap, in this paper, we firstly theoretically demonstrate that the optimization objectives when incorporating group MMF constraint can be essentially reformulated as a group-weighted accuracy optimization objective on different groups. Then, we introduce a large-scale friendly, and effective algorithm called FairDual to optimize the group-weighted objective for minimizing the Jensen gap. Specifically, we formulate the fairness-constraint problem as its dual, where the dual variable (referred to as the shadow price in economics (Drèze and Stern, 1990)) can be interpreted as the sample weight assigned to each sample in the mini-batch optimization process. Then, FairDual leverages dual-optimization techniques to optimize the weight of different group losses utilizing dual mirror gradient techniques efficiently.

Our theoretical analysis demonstrates that FairDual can achieve a sub-linear convergence rate to the globally optimal point under a random shuffling mini-batch training style. Moreover, the Jensen gap can be well bounded (See Section 5.2.3) even when confronted with small mini-batch sizes and large group sizes. Extensive experiments using six large-scale RS backbone models on three publicly available datasets show that FairDual consistently reduces the Jensen gap and outperforms all baselines with a large margin in terms of both accuracy and fairness while achieving better efficiency.

## 2 RELATED WORK

**Fairness concept in RS.** One common categorization is based on the involvement of different stakeholders (Abdollahpouri et al., 2020; Abdollahpouri and Burke, 2019), divides fairness into individual fairness (Marras et al., 2022; Li et al., 2021), which aims to ensure equitable treatment for individual users, and group fairness, which classifies items into various groups (Xu et al., 2023; 2024b; Patro et al., 2020; Naghiaei et al., 2022; Wu et al., 2021). Various approaches have been proposed to optimize fairness utilizing different fairness objectives. For instance, Patro et al. (2020) proposed using the Shapley value, while Do and Usunier (2022) suggested optimizing the Gini Index. On the other hand, works such as Xu et al. (2023); Do et al. (2021); Xu et al. (2024b) advocate for optimizing MMF, which requires every group should receive a "minimum wage". Typically, we mainly focus on the group MMF, which is closer to the industrial scenarios since certain studies propose to ensure minimum item exposures for attracting providers to join or enhancing the visibility of specific item categories (Patro et al., 2020; Xu et al., 2024b; Zhu et al., 2020).

**Optimizing fairness in RS.** When optimizing fairness, previous research often categorizes methods into three categories based on recommendation phases, including pre-processing (Calmon et al.,

2017; Xiong et al., 2024; Xu et al., 2024a), post-processing (Xu et al., 2023; Patro et al., 2020; Wu et al., 2021; Xu et al., 2024c) and in-processing (Narasimhan et al., 2020; Tang et al., 2023). In this paper, we theoretically demonstrate that the in-processing method constrained by group MMF can be essentially reformulated as a re-weighting approach. Prior research employed static or dynamic group weights to achieve fairness. For static weights, Jiang et al. (2024); Xiong et al. (2024) proposes to set item weight according to its popularity and the Wasserstein distance of two groups, respectively. For dynamic weighting, some work (Chen et al., 2023b; Chai and Wang, 2022; Wen et al., 2022) propose to design weights based on the training state, while Hu et al. (2023) employs a dynamic re-weighting strategy to mitigate distribution shifts between training and test data. Roh et al. (2020) also proposes to set different batch sizes to optimize fairness. However, these methods are either designed for simple cases involving only two groups, or they lack theoretical guarantees when applied to group MMF settings.

**Optimizing fairness in ML.** In machine learning (ML), previous work aims to optimize different fairness functions to achieve various social welfare objectives. For example, the power-mean welfare family seeks to balance accuracy and fairness objectives by applying the exponential form (Cousins, 2021; 2023) and max-min fairness (Abernethy et al., 2022; Agarwal et al., 2018) aims to support the worst-off groups. When optimizing fairness, we commonly try to optimize a nonlinear fairness function. When adopting optimization methods such as stochastic gradient descent (SGD), an unavoidable bias will exist (Demidovich et al., 2023; Hu et al., 2020). To bridge this bias, previous ML methods have employed sampling strategies (Abernethy et al., 2022; Cousins, 2022) to obtain unbiased samples, while some methods have utilized debiasing SGD (Demidovich et al., 2023; Agarwal et al., 2018) to mitigate the bias. However, these works cannot be applied to large-scale industrial RS since they often require a convex optimization process that is impractical for RS tasked with serving millions of users and hundreds of groups. To efficiently bridge the Jensen gap, our method improves debiasing SGD by developing a large-scale friendly mirror SGD learning algorithm.

## 3 FORMULATION

In RS, let $\mathcal{U}, \mathcal{I}$ be the set of users and items, and each item $i \in \mathcal{I}$ is associated with a unique group $g \in \mathcal{G}$, where the set of items associated with $g$ is denoted as $\mathcal{I}_g$. In RS, an item $i$ may belong to a different group $g$ (*e.g.,* a movie can be categorized under various genres such as action, or drama). We define the number of groups to which the item $i$ belongs as $n_i$.

Suppose that the RS manages a set of user-item historical interactions $\mathcal{D} = \{(u, i, c_{u,i})\}$, where each tuple $(u, i, c_{u,i})$ records that a user $u \in \mathcal{U}$ accessed the system and interacted with an item $i \in \mathcal{I}$ with behavior $c_{u,i} \in \{0, 1\}$. $c_{u,i} = 1$ means that the user $u$ clicked/purchased the item $i$, and 0 otherwise. The task of recommendation becomes, based on the user-item interactions in $\mathcal{D}$, learning an empirical estimation $\hat{c}_{u,i} = f(u, i)$ for real label $c_{u,i}$. Then RS will suggest $K$ items to the user according to predicted preference scores $\hat{c}_{u,i}$, with the ranking list denoted as $L_K(u) \in \mathcal{I}^K$. In general, the $f(\cdot)$ can be either the traditional matrix factorization model (He et al., 2016) or more advanced LLMs-based recommender models (Bao et al., 2023).

Following the practice in recommendation tasks (He et al., 2017; 2016), the cross-entropy loss $-c_{u,i} \log(\hat{c}_{u,i})$ is regarded as a common and better choice compared to other loss functions. Meanwhile, to fulfill the group MMF requirement (Patro et al., 2020; Xu et al., 2024b), the recommendation model also strives to maintain the expected utility of a specific group $g$ (where the group's utility is defined as the negative sum of the entropy loss within the group) exceeds a basic threshold $M$. MMF constraint aims to ensure every group can receive the required group-specific "minimum wage" during the training phases. Formally, we can write the ideal optimization objective as follows:

$$\mathcal{L} = \min_{\hat{c}_{u,i}} \underbrace{- \sum_{u \in \mathcal{U}} \sum_{i \in \mathcal{I}} c_{u,i} \log(\hat{c}_{u,i})}_{\text{recommendation accuracy loss}} \quad \text{s.t.} \quad \underbrace{\max_{g \in \mathcal{G}} \sum_{u \in \mathcal{U}} \sum_{i \in L_K(u)} -\frac{\mathbb{I}(i \in \mathcal{I}_g)}{n_i m_g} c_{u,i} \log(\hat{c}_{u,i}) \leq M}_{\text{MMF constraint: loss of worst-off group } g \text{ should at or smaller than } M} , \quad (1)$$

where $\mathbb{I}(\cdot)$ denotes the indicator function, and the number of users $|\mathcal{U}|$ could represent the daily or weekly user traffic. The $m_g$ can be regarded as the weight for different group $g$. Note that, following the practice in time-aware RS (Kang and McAuley, 2018b; Sun et al., 2019), we utilize the recent $H$ interactions, which represent the truncated user historical behavior numbers.

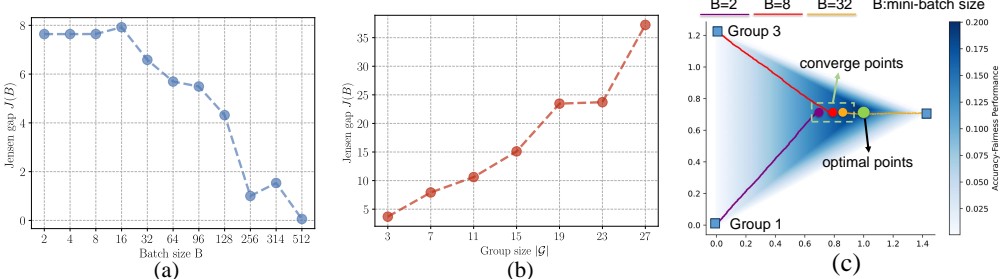

Figure 1: Loss converges simulation with 1000 users and 1000 items. Sub-figure (a) and (b) illustrate the distance away from the optimal point (*i.e.*, Jensen gap) *w.r.t.* mini-batch and group size, respectively. Figure (a) was conducted with the same group size (G=7) under different batch sizes, while Figure (b) was conducted with the same batch size (B=32) under different group sizes. Sub-figure (c) describes the converged trace under different batch sizes.

## 4 PROBLEM ANALYSIS

In real-world scenarios, the number of users $|\mathcal{U}|$ is often large, and a mini-batch sampling strategy with a batch size of B is often necessary due to the large computational costs. Each batch only contains a subset of users. However, we show that the non-linear additivity property of the MMF-based objective will introduce the Jensen gap between the model's convergence point and the optimal point when employing mini-batch sampling strategies.

In this section, we analyze why the Jensen gap exists and how it will influence the model's convergence in both theoretical and empirical ways.

### 4.1 THEORETICAL ANALYSIS

Firstly, we will re-write the optimization objective using the following theorem:

**Theorem 1.** *For a vector $\boldsymbol{x} \in \mathbb{R}^n$, $\boldsymbol{x}^i$ denotes the element of the vector raised to the power of $i$. Similarly, $\log(\boldsymbol{x})$ denotes the element of the vector reduced as $\log(\boldsymbol{x}_i)$. Let $\boldsymbol{A} \in \mathbb{R}^{|\mathcal{I}| \times |\mathcal{G}|}$ is the item-group adjacent matrix, and $\boldsymbol{A}_{ig} = 1$ indicates item $i \in \mathcal{I}_g$, and 0 otherwise. Let $\boldsymbol{w} \in \mathbb{R}^{|\mathcal{I}|} = [-\sum_{u \in \mathcal{U}} c_{u,i} \log(\hat{c}_{u,i})]_{i \in \mathcal{I}}$ and its feasible region is $\mathcal{W} = \{\boldsymbol{w} | \sum_{i \in \mathcal{I}} c_{u,i} \leq K, \forall u \in \mathcal{U}, c_{u,i} \in [0,1]\}$. Then there exist $t \in [0, \infty)$ (value of $t$ relates to the value of $M$) and a weight vector $\boldsymbol{b} \in \mathbb{R}^{|\mathcal{G}|} \geq 0$, s.t. Equation (1) can be optimized as:*

$$\mathcal{L} = \min_{w \in \mathcal{W}} \boldsymbol{b}^\top (\hat{\boldsymbol{A}}^\top \boldsymbol{w})^{1+t} \tag{2}$$

*where $\hat{\boldsymbol{A}}$ is the row-normalized matrix for $\boldsymbol{A}$: $\hat{\boldsymbol{A}} = diag(\boldsymbol{A}\boldsymbol{1})^{-1}\boldsymbol{A}$. $diag(\boldsymbol{x})$ denotes to construct a diagonal matrix based on vector $\boldsymbol{x}$.*

The detailed proof of Theorem 1 can be seen in Appendix A. When transforming the original optimization process in Equation (1) into Equation (2), we can easily observe that the loss function does not adhere to linear additivity. Then the Jensen gap will arise under mini-batch sample strategy by formulating it using the following theorem:

**Theorem 2.** *Under mini-batch sample strategies, we partition the user set $\mathcal{U}$ into $|\mathcal{U}|/B$ subsets and perform optimization on each subset. Let $\boldsymbol{e}^j \in \mathbb{R}^{|\mathcal{G}|}$ be the group accumulated utility under $j$-th partition, where each element $\boldsymbol{e}_{j,g} = -\sum_{u \in \mathcal{U}_j} \sum_{i \in \mathcal{I}_g} c_{u,i} \log(\hat{c}_{u,i})$. Let $f(\boldsymbol{x}) = x^{t+1}$. We can write the mini-batch optimizing loss objective $\mathcal{L}^B$ as: $\mathcal{L}^B = \min \sum_{j=1}^{|\mathcal{U}|/B} \boldsymbol{b}^\top f(\boldsymbol{e}_j)$, where $\mathcal{U}_j$ is the $j$-th partition of the user set $\mathcal{U}$. Then, Jensen gap (Gao et al., 2017; Ullah et al., 2021) is defined as:*

$$J(B) = |\mathcal{L}^B - \mathcal{L}| = |\mathcal{L}^B - \min \boldsymbol{b}^\top f(\sum_{j=1}^{|\mathcal{U}|/B} \boldsymbol{e}_j)| \neq 0. \tag{3}$$

*When optimizing Equation (2) under the mini-batch sampling style, the mini-batch size $B$ becomes smaller and group size $\mathcal{G}$ becomes larger, the Jensen gap $J(B)$ will become larger. Moreover, when*

*the mini-batch size becomes smaller, we are more likely to underestimate the original loss,* i.e., $\mathcal{L}^B \leq \mathcal{L}$. *The loss underestimation will result in the Jensen gap.*

The detailed proof of Theorem 2 can be seen in Appendix C. The intuitive reason behind the Jensen gap raised by group MMF is that the accuracy-fairness trade-off problem does not adhere to linear additive attributes. Essentially, the combination of different batches forms a concave function. Mini-batch size and group size both measure the degree of data partitioning, where smaller batch sizes and larger group sizes lead to fewer data partitions. As a result, due to the non-linear and intricate function form of a neural network (Sun, 2019), these errors in estimating the loss function impede the model from converging to the optimal point, thus diminishing the performance. Next, we will give an empirical analysis to prove this.

## 4.2 EMPIRICAL ANALYSIS

In this section, we illustrate a simulation (Figure 1) conducted under the assumption of knowing every user-item true preference score to validate the correctness of our theoretical analysis. We use the simple recommendation model: Matrix Factorization (Singh and Gordon, 2008) since we can have a closed-form expression on parameter updating. Then we apply a common mini-batch training strategy to optimize accuracy-fairness objective based on the parameters outlined in Xu et al. (2023); Patro et al. (2020), with the accuracy-fairness coefficient of 0.5.

As shown in Figure 1 (a) and (b), we uncover that the Jensen gap (distance away from the optimal point) will deviate with smaller mini-batch sizes and larger group sizes. Figure (c) describes the converge trace under different batch sizes by mapping the top-K simplex space of three groups of recommendation ranking to a 2-dimensional space through a topological homeomorphic transformation (Kozlov, 2008). Figure 1 (c) also indicates that different batch sizes result in different gradient optimization directions, with smaller batch sizes leading to larger shifts in the error of the optimization direction. These empirical results confirm the correctness of our theoretical analysis.

For other types of fairness, such as the power-mean welfare family (Cousins, 2021) and the Gini welfare function (Do and Usunier, 2022), also exhibit non-linear properties, leading to analogous Jensen gap phenomena. We discuss them in the Appendix H and Appendix M.

## 5 METHOD

In this section, we will introduce our method FairDual.

### 5.1 OPTIMIZING MAX-MIN FAIRNESS AS GROUP-WEIGHTED OBJECTIVE

In this section, in order to tackle this problem, we show that the MMF-constrained objective can be regarded as the group-weighted optimization problem using the following theorem:

**Theorem 3.** *By introducing the dual variable $\boldsymbol{\mu}$, the dual form of the Equation (1) is*

$$\mathcal{L}' = \min_{\hat{c}_{u,i}} \quad -\sum_{u \in \mathcal{U}} \sum_{g \in \mathcal{G}} \boldsymbol{s}_g \sum_{i \in \mathcal{I}_g} c_{u,i} \log(\hat{c}_{u,i}), \tag{4}$$

*where $\boldsymbol{s}_g = 1 - \boldsymbol{\mu}_g$ and $\boldsymbol{\mu} = \arg\min_{\boldsymbol{\mu} \in \mathcal{M}} \left( \max \sum_{u \in \mathcal{U}} \sum_{g \in \mathcal{G}} \boldsymbol{s}_g \sum_{i \in \mathcal{I}_g} c_{u,i} \log(\hat{c}_{u,i}) + \lambda r^*(\boldsymbol{\mu}) \right)$, where $r^*(\mu) = \max_{\boldsymbol{w}_g \leq m_g} \left( \min_{g \in \mathcal{G}} m_g(\hat{A}\boldsymbol{w})_g + \hat{A}^\top \boldsymbol{w}\boldsymbol{\mu}/\lambda \right) = \sum_g m_g \boldsymbol{\mu}_g/\lambda + 1$, $\mathcal{M} = \left\{ \boldsymbol{\mu} \,\middle|\, \sum_{g \in \mathcal{S}} \boldsymbol{\mu}_g m_g \geq -\lambda, \forall \mathcal{S} \in \mathcal{G}_s \right\}$, where $\mathcal{G}_s$ is the set of all subsets of $\mathcal{G}$ (i.e., power set).*

The detailed proof of Theorem 3 can be seen in Appendix F. From Theorem 3, we observe that the recommendation task constrained by max-min fairness can be viewed as a re-weighting approach across different groups on the original loss function solely optimized for accuracy.

Intuitively, $s_g = 1 - \mu_g$ is the negative shadow price (Drèze and Stern, 1990). The high shadow price $\mu_g$ indicates that this constraint is the primary factor constraining accuracy optimization. Conversely, a low or zero shadow price suggests that the fairness constraint currently imposes little restriction on accuracy. Specifically, a high $s_g$ signifies that this constraint is the primary factor limiting fairness

---

**Algorithm 1:** FairDual

---

**Require:** Dataset $\mathcal{D} = \{u, i, c_{u,i}\}$, item-group adjacent matrix $\boldsymbol{A}$, dual learning rate $\eta$, trade-off coefficient $\lambda$, $m^i_{\text{freeze}}(\cdot)$ updating step $\beta$, batch size $B$ and sample item number $Q$ and the weight $m_g$ for each group $g$. $\hat{\boldsymbol{A}} = \text{diag}(\boldsymbol{A1})^{-1}\boldsymbol{A}$.

**Ensure:** The model parameters of $m^i(\cdot), m^u(\cdot)$.

1: **for** $n = 1, \cdots, N$ **do**
2:     Set $\boldsymbol{\gamma}_{1,g} = m_g, \forall g \in \mathcal{G}$
3:     **for** $j = 1, \cdots, |\mathcal{U}|/B$ **do**
4:         **if** $(n * |\mathcal{N}|/B + j)\%\beta = 0$ **then**
5:             Copy parameters from $m^i(\cdot)$ to $m^i_{\text{freeze}}(\cdot)$ and get all item embedding $\boldsymbol{E}$
6:             Initialize dual solution $\boldsymbol{\mu} = 0$, and momentum gradient $\boldsymbol{g} = 0$ and $t = 0$.
7:         **end if**
8:         Get sub-dataset $\{u, i, c_{u,i}\}^B_{b=1}$ and user feature $\boldsymbol{e}_u$ and item feature $\boldsymbol{e}_i$
9:         $\mathcal{L}^j = [-c_{u,i}\log(\hat{c}_{u,i})]^B_{b=1}, \quad \boldsymbol{s}^j = \boldsymbol{1} - \hat{\boldsymbol{A}}^j\boldsymbol{\mu}$
10:        Apply gradient descent based on loss $(\boldsymbol{s}^j)^\top\mathcal{L}^j$
11:        $\widetilde{\boldsymbol{w}}_b = \sum^K_{k=1}(\boldsymbol{e}^\top_{u_b}\boldsymbol{E}^b)_{[k]}, \forall b$
12:        $\widetilde{\boldsymbol{g}}^j = -(\hat{\boldsymbol{A}}^j)^\top\widetilde{\boldsymbol{w}} + \boldsymbol{\gamma}_j \quad , \boldsymbol{g}^j = \alpha\widetilde{\boldsymbol{g}}^j + (1-\alpha)\boldsymbol{g}, \quad \boldsymbol{g} = \boldsymbol{g}^j$
13:        $\boldsymbol{\gamma}_j = \boldsymbol{\gamma}_{j-1} - (\hat{\boldsymbol{A}}^j)^\top\widetilde{\boldsymbol{w}}, \boldsymbol{\mu} = \arg\min_{\boldsymbol{\mu}_0\in\mathcal{M}}\left[(\boldsymbol{g}^j)^\top\boldsymbol{\mu}_0 + \eta\|\boldsymbol{\mu}_0 - \boldsymbol{\mu}\|^2_2\right]$
14:     **end for**
15: **end for**

---

optimization for group $g$, whereas a low or zero $s_g$ implies that the accuracy constraint for group $g$ currently has little impact on the overall optimization.

## 5.2 FAIRDUAL

We then will introduce our method FairDual under random shuffling mini-batch training strategies. The overall workflow of FairDual under every two batches $j$ and $j + 1$ can be seen in Figure 2. According to analysis in Theorem 3, under each epoch, the overall optimization process will become:

$$\mathcal{L}'^B = \min \sum_{j=1}^{|\mathcal{U}|/B} (\boldsymbol{s}^j)^\top\boldsymbol{l}^j, \tag{5}$$

where $\boldsymbol{l}^j \in \mathbb{R}^B, \boldsymbol{s}^j \in \mathbb{R}^B$ is loss and its weight under $j$-th batch. Next, we will explain how $\boldsymbol{l}^j$ and $\boldsymbol{s}^j$ update on each batch $j$. Detailed algorithm workflow can be seen in Algorithm 1. Note that, following the practice in Bao et al. (2023), we utilize the user's historical behaviors to represent each user, thereby treating each sample as a unique user.

### 5.2.1 ACCURACY LOSS CONSTRUCTING

In the mainstream recommender architectures, the primary objective is to make the predicted score close to the true user preference. That is, at each batch $j$, there are $B$ user-item pair $[(u, i)]^B_{b=1}$ arrives. Then the loss vector $\mathcal{L}^j$ is computed as:

$$\boldsymbol{l}^j = [-c_{u,i}\log(\hat{c}_{u,i})]^B_{b=1}, \tag{6}$$

where $\hat{c}_{u,i} = -d(\boldsymbol{e}_u, \boldsymbol{e}_i) \leq 1$, where $d(\cdot)$ is the normalized distance between embedding $\boldsymbol{e}_u, \boldsymbol{e}_i$. The commonly used distance metric is the dot-product, i.e. $d(\boldsymbol{e}_u, \boldsymbol{e}_i) = -\boldsymbol{e}^\top_u\boldsymbol{e}_i$, and the $\boldsymbol{e}_u$ and $\boldsymbol{e}_i$ are calculated by a complex model, i.e. $\boldsymbol{e}_u = m^u(u), \boldsymbol{e}_i = m^i(i)$, where $m^u(\cdot)$ and $m^i(\cdot)$ are two embedding extraction networks. Typically, the user $u$ is represented by the item-clicked history sequences before clicking the item $i$: $[i^1, i^2, \cdots, i^n]$, where $n$ is the fixed item sequence length.

Note that in text-based recommendation models such as BigRec (Bao et al., 2023) and Recformer (Li et al., 2023), each item $i$ is represented as a sequence of words in natural language: $i = [w^1, w^2, \cdots, w^l]$, where $l$ is the length of the sentence and user behaviors are also represented in prompt form (Bao et al., 2023). In such cases, Equation (6) is extended to

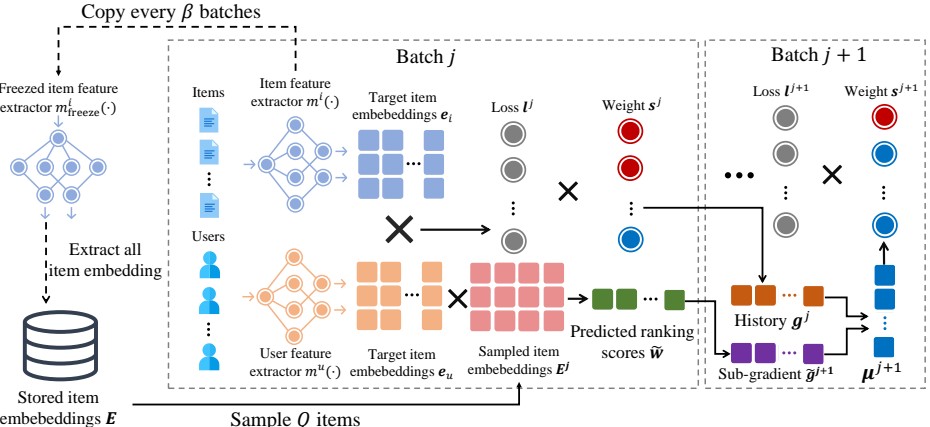

Figure 2: Overall workflow of FairDual under every two batches $j$ and $j + 1$.

$\log \hat{c}_{u,i} = \sum_{i=1}^{l} \log(P(w^i))$, where $P(w)$ refers to the predicted probability of the word $w$ generated by the LLMs, while other equations remain unchanged.

### 5.2.2 MIRROR GRADIENT DECENT FOR GROUP WEIGHT

For each batch $j$, the model needs to decide the weight $s^j$ for each sample. The weight $s^j$ is computed utilizing mirror-gradient descent (Balseiro et al., 2021) technique. Specifically,

$$s^j = \mathbf{1} - \hat{A}^j \mu^j, \tag{7}$$

where $\hat{A}^j \in \mathbb{R}^{B \times |\mathcal{G}|}$ represents the row-normalized item-group adjacency matrix for $A$ in batch $j$ (see details in Equation (2)), with $A_{i_b,g} = 1$ indicating that the $b$-th item in batch $i_b \in \mathcal{I}_g$ belongs to group $g$, and 0 otherwise. And $\mu^j$ is the dual variable at $j$-th batch, which updates as:

$$\mu^j = \arg\min_{\mu} \left[ (g^j)^\top \mu + \eta \| \mu - \mu^{j-1} \| \right], \quad \text{s.t.} \sum_{j=1}^{g} \mu_j m_j + \lambda \geq 0, \ \forall g = 1, 2, \ldots, |\mathcal{G}|, \tag{8}$$

where $\eta$ is the learning rate, $g^j$ is the sub-gradient of the Equation (4) *w.r.t.* the dual variable $\mu^j \in \mathbb{R}^{|\mathcal{G}|}$. The projection step can be efficiently solved using convex optimization solvers (Balseiro et al., 2021) since $\mathcal{D}$ is coordinate-wisely symmetric.

Specifically, to ensure smoothness and make user of historical information, we utilize the momentum gradient descent to update $g^j$:

$$g^j = \alpha \tilde{g}^j + (1 - \alpha) \mathbf{g}^{j-1}, \quad \tilde{g}^j = \partial(s^j \mathcal{L}^j + \lambda r^*(\mu^j)) = -(\mathbf{A}^j)^\top \tilde{w} + \gamma_j, \tag{9}$$

where $\gamma_j \in \mathbb{R}^{|\mathcal{G}|}$ is the vector, whose element of index $g$ denotes the remaining required loss (*i.e.*, reward) for the group $g$ at batch step $j$, $\tilde{w} \in \mathbb{R}^B$ represents the estimated ranking score that each user query will receive. However, given the vast size of the item corpus in recommendation systems, conducting a full ranking on all items is impractical. Therefore, we randomly sample $Q$ items to approximate the ranking scores across all items. The $Q$ items' embeddings are denoted as $E^j \in \mathbb{R}^{Q \times d}$. Formally, for the $b$-th element $\tilde{w}_b$, we can write: $\tilde{w}_b = \sum_{k=1}^{K} (E^j e_{u_b})_{[k]}$, where $x_{[k]}$ denote the $k$-th largest element in vector $x$ and $K$ is the ranking size.

Note that $E^b$ is sampled from the pre-stored item embedding $E \in \mathbb{R}^{|\mathcal{I}| \times d}$, which is pre-calculated using the freezer network $m^i_{\text{freeze}}(\cdot)$. This is done to mitigate the significant fluctuations in $\tilde{w}$ caused by unstable training (Fan et al., 2020). To achieve this, we freeze the item feature extractor $m^i(\cdot)$ as $m^i_{\text{freeze}}(\cdot)$ and transfer the parameters from $m^i(\cdot)$ to $m^i_{\text{freeze}}(\cdot)$ every $\beta$ batches. For the first batch process, we initialize $g^1$ as $\mathbf{0}$, which will not make an effect on the first batch.

### 5.2.3 BOUND ON JENSEN GAP

We will provide the Jensen gap converge analysis of FairDual in the following theorem.

**Theorem 4** (Bound on Jensen Gap). *There exists $H > 0$ such that $\|\boldsymbol{\mu}^j - \boldsymbol{\mu}\|_2^2 \leq H$ and function $\|\cdot\|_2^2$ is $\sigma-$strongly convex. Then, there exists $L > 0$, the Jensen gap of FairDual can be bounded as:*

$$J(B) \leq \frac{H}{\eta} + \frac{|\mathcal{U}|L|\mathcal{G}|^2}{B(1-\alpha)\sigma}\eta + \frac{L|\mathcal{G}|^2}{2(1-\alpha)^2\sigma\eta}. \tag{10}$$

*When setting learning rate $\eta = O(B^{-1/2})$, the bound of Jensen gap is comparable with $O(B^{-1/2})$.*

The detailed proof can be seen in Appendix G. From Theorem 4, it is apparent that the Jensen gap will widen as the batch size $B$ decreases and the group size $|\mathcal{G}|$, as well as the max-min fairness degree $\lambda$, increase. However, FairDual demonstrates a sub-linear convergence rate concerning the batch size $B$, and it maintains strong performance even with small batch sizes and large group sizes across various fairness degrees.

## 6 EXPERIMENT

We conduct experiments to demonstrate the effectiveness of the proposed FairDual.

### 6.1 EXPERIMENTAL SETTINGS

**Datasets**. The experiments are conducted on the commonly used two widely used and publicly available recommendation datasets, including MIND (Wu et al., 2020)[1], Amazon-Book and Amazon-Electronic (He and McAuley, 2016)[2]. Their detailed statistical information is in Appendix I.

**Evaluation**. We arrange all interactions in the dataset chronologically by their timestamps and employ the first 80% interactions as training data. The remaining 20% of interactions are divided equally, with each 10% segment used for validation and testing, respectively, during evaluation.

Regarding the metric, following the practice in Dai et al. (2023), we utilize Normalized Discounted Cumulative Gain (NDCG) and mean Reciprocal Rank (MRR) to measure the accuracy: NDCG@K $= \frac{1}{|\mathcal{U}|}\sum_{u=1}^{|\mathcal{U}|}\frac{\sum_{i \in L_K(u)}(2^{c_{u,i}}-1)/(\log_2(j+1))}{(2^{\mathrm{rank}_i}-1)/(\log_2(\mathrm{rank}_i+1))}$, MRR@K $= \frac{1}{|\mathcal{U}|}\sum_{u=1}^{|\mathcal{U}|}\frac{1}{\mathrm{rank}_i}$, where $\mathrm{rank}_i$ is the rank of the first correct answer. Meanwhile, we employ MMF@K to gauge the degree of fairness, which quantifies the aggregated ranking score of the 20% worst-off groups (Do et al., 2021; Xu et al., 2023).

**Backbones and baselines**. For the backbone, we first select three large-scale recommender models: **NRMS** (Wu et al., 2019), **RecFormer** (Li et al., 2023) and **BigRec** (Bao et al., 2023). Note that BigRec only utilizes 1024 samples to train due to large computational cost.

For the baselines, we choose several fair-aware re-weight baselines that aim to improve group MMF: **UNI** (without considering fairness), **DRO** (Hashimoto et al., 2018), **S-DRO** (Wen et al., 2022), **Prop** (Hu et al., 2023), **IFairLRS** (Jiang et al., 2024) and **Maxmin Sample** (Abernethy et al., 2022).

The detailed descriptions of the backbones and baselines are in Appendix I.

**Implemtation details.** We provide our detailed running environment, all hyper-parameter settings, utilized LLMs settings, and used the toolkit in Appendix I.

### 6.2 EXPERIMENTAL RESULTS ON FULL DATASETS

Firstly, we conduct experiments to show the performance of FairDual and other baselines across all large-scale recommendation backbones. Table 1 presents the experimental outcomes for our FairDual model and the baseline methods across all datasets, respectively. Table 2 presents the experimental outcomes for our FairDual model and the baseline methods across other different backbones on the MIND dataset. To make fair comparisons, all the baselines were tuned to their hyperparameters to obtain the best trade-off accuracy and fairness performance under our settings.

From the experiments, it is evident that FairDual consistently outperforms the baseline methods across all datasets and various base models, spanning different top-K ranking sizes. This is reflected

---

[1]https://microsoftnews.msn.com
[2]http://jmcauley.ucsd.edu/data/amazon/

Table 1: Performance comparisons between ours and the baselines on three datasets under best-performing BigRec backbones. The $*$ means the improvements are statistically significant (t-tests and $p$-value $< 0.05$). The bold number indicates that the accuracy value exceeds that of all the baselines.

| Models/Metrics | | $K = 5$ | | | $K = 10$ | | | $K = 20$ | | |
|---|---|---|---|---|---|---|---|---|---|---|
| | | NDCG (%) | MRR (%) | MMF (%) | NDCG (%) | MRR (%) | MMF (%) | NDCG (%) | MRR (%) | MMF (%) |
| **MIND** | UNI | 1.02 | 0.79 | 1.63 | 1.50 | 0.98 | 2.33 | 2.16 | 1.16 | 2.94 |
| | DRO | 0.90 | 0.67 | 1.81 | 1.37 | 0.87 | 2.51 | 1.94 | 1.02 | 3.21 |
| | Prop | 1.11 | 0.88 | 1.97 | 1.62 | 1.09 | 2.53 | 2.14 | 1.23 | 3.05 |
| | S-DRO | 0.91 | 0.70 | 1.87 | 1.42 | 0.91 | 2.41 | 1.93 | 1.04 | 3.02 |
| | IFairLRS | 0.87 | 0.66 | 2.21 | 1.27 | 0.83 | 2.91 | 1.78 | 0.97 | 2.86 |
| | Maxmin sample | 0.98 | 0.75 | 2.25 | 1.49 | 0.96 | 1.71 | 2.19 | 1.15 | 3.13 |
| | **Ours** | **1.15**$^*$ | **0.88** | **2.82**$^*$ | **1.69**$^*$ | **1.11** | **2.99**$^*$ | **2.28**$^*$ | **1.27**$^*$ | **3.39**$^*$ |
| | improv.(%) | 3.60 | 0.00 | 25.33 | 4.32 | 1.83 | 2.75 | 4.10 | 3.25 | 5.61 |
| **Book** | UNI | 2.99 | 2.79 | 8.44 | 3.19 | 2.87 | 8.32 | 3.44 | 2.94 | 8.15 |
| | DRO | 2.94 | 2.72 | 8.39 | 3.15 | 2.81 | 8.29 | 3.37 | 2.87 | 8.10 |
| | Prop | 2.64 | 2.45 | 8.68 | 2.83 | 2.53 | 8.30 | 3.05 | 2.59 | 8.01 |
| | S-DRO | 2.61 | 2.44 | 8.37 | 2.80 | 2.52 | 8.21 | 3.06 | 2.59 | 8.07 |
| | IFairLRS | 2.30 | 2.16 | 8.46 | 2.51 | 2.25 | 8.20 | 2.76 | 2.32 | 8.17 |
| | Maxmin sample | 2.49 | 2.31 | 6.80 | 2.72 | 2.43 | 6.80 | 2.97 | 2.74 | 7.50 |
| | **Ours** | **3.11**$^*$ | **2.88** | **8.90**$^*$ | **3.31**$^*$ | **2.96** | **9.00**$^*$ | **3.60**$^*$ | **3.04** | **8.89**$^*$ |
| | improv.(%) | 4.01 | 3.23 | 2.53 | 3.76 | 3.14 | 8.17 | 4.65 | 3.40 | 8.81 |
| **Electronic** | UNI | 4.61 | 4.30 | 0.26 | 4.93 | 4.43 | 0.25 | 5.30 | 4.53 | 0.21 |
| | DRO | 4.65 | 4.34 | 0.24 | 4.96 | 4.46 | 0.24 | 5.33 | 4.57 | 0.21 |
| | Prop | 4.63 | 4.33 | 0.26 | 4.96 | 4.47 | 0.25 | 5.33 | 4.57 | 0.21 |
| | S-DRO | 4.60 | 4.29 | 0.25 | 4.92 | 4.42 | 0.24 | 5.29 | 4.52 | 0.20 |
| | IFairLRS | 2.21 | 2.06 | 0.19 | 2.46 | 2.16 | 0.17 | 2.69 | 2.22 | 0.12 |
| | Maxmin sample | 4.60 | 4.31 | 0.27 | 4.92 | 4.44 | 0.25 | 5.31 | 4.55 | 0.21 |
| | **Ours** | **5.08**$^*$ | **4.78** | **0.31**$^*$ | **5.43**$^*$ | **4.92** | **0.30**$^*$ | **5.84**$^*$ | **5.03** | **0.26**$^*$ |
| | improv.(%) | 9.24 | 10.1 | 14.8 | 9.47 | 10.0 | 19.9 | 0.95 | 10.0 | 23.8 |

Table 2: Performance comparisons between ours under other backbones on MIND dataset. The $*$ means the improvements are statistically significant (t-tests and $p$-value $< 0.05$). The bold number indicates that the accuracy value exceeds that of all the baselines.

| Models/Metrics | | top-5 | | | top-10 | | | top-20 | | |
|---|---|---|---|---|---|---|---|---|---|---|
| | | NDCG (%) | MRR (%) | MMF (%) | NDCG (%) | MRR (%) | MMF (%) | NDCG (%) | MRR (%) | MMF (%) |
| **NRMS** | DRO | 0.44 | 0.32 | 0.12 | 0.66 | 0.42 | 3.60 | 1.06 | 0.50 | 9.94 |
| | Prop | 0.44 | 0.32 | 0.12 | 0.66 | 0.42 | 3.49 | 1.06 | 0.52 | 9.94 |
| | S-DRO | 0.52 | 0.34 | 0.10 | 0.76 | 0.40 | 2.05 | 1.20 | 0.52 | 8.74 |
| | IFairLRS | 0.40 | 0.28 | 0.69 | 0.62 | 0.36 | 4.20 | 0.96 | 0.44 | 10.58 |
| | Maxmin sample | 0.38 | 0.31 | 0.20 | 0.45 | 0.34 | 4.00 | 0.67 | 0.422 | 9.99 |
| | **Ours** | **0.60**$^*$ | **0.40**$^*$ | **1.07**$^*$ | **0.84**$^*$ | **0.46**$^*$ | **4.93**$^*$ | **1.28**$^*$ | **0.60**$^*$ | **11.35**$^*$ |
| **RecFormer** | DRO | 0.57 | 0.45 | 1.08 | 0.89 | 0.59 | 1.08 | 1.41 | 0.73 | 1.52 |
| | Prop | 0.57 | 0.45 | 1.08 | 0.89 | 0.58 | 1.08 | 1.41 | 0.72 | 1.52 |
| | S-DRO | 0.57 | 0.45 | 1.20 | 0.91 | 0.60 | 1.15 | 1.46 | 0.73 | 1.62 |
| | IFairLRS | 0.46 | 0.37 | 1.68 | 0.76 | 0.49 | 1.70 | 1.29 | 0.63 | 2.12 |
| | Maxmin sample | 0.51 | 0.41 | 0.94 | 0.85 | 0.55 | 1.50 | 1.37 | 0.69 | 2.48 |
| | **Ours** | **0.59**$^*$ | **0.45** | **1.88**$^*$ | **0.99**$^*$ | **0.60** | **1.94**$^*$ | **1.55**$^*$ | **0.75** | **2.58**$^*$ |

in accuracy metrics such as NDCG and MRR, as well as the fairness metric MMF. The results conclusively demonstrate that FairDual effectively ensures the model reaches a better convergence point in terms of both accuracy and fairness by leveraging dual gradient descent.

We also conduct experiments on traditional RS backbones, as detailed in Appendix J. The results also consistently confirm the effectiveness of our model FairDual.

## 6.3 EXPERIMENTAL ANALYSIS

In this section, we first replicate the simulation settings outlined in Section 4.2 to investigate how the Jensen gap changes. Then we conduct analysis on MIND dataset under BigRec base models.

**Jensen gap.** Firstly, we investigate the variations in the Jensen gap concerning batch size $B$ and group size $|\mathcal{G}|$ across both baseline methods and our proposed FairDual model. As shown in Figure 3 (a), we can see that FairDual has a lower Jensen gap than other online models across different batch sizes and group sizes. Furthermore, it's evident that the Jensen gap exhibited by FairDual remains consistently stable across various batch sizes, with only a marginal increase observed as the group size expands. This indicates that FairDual can consistently maintain a low Jensen gap level.

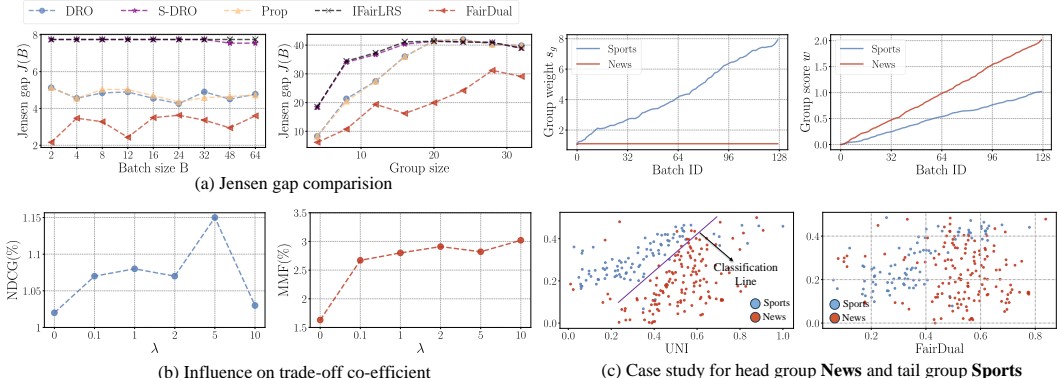

Figure 3: Sub-figure (a) conducts a simulation to show Jensen gap $J(B)$ changes *w.r.t.* batch size $B$ and group size $|\mathcal{G}|$ for all baselines and FairDual. Sub-figure(b,c) conducts on MIND dataset under BigRec. Sub-figure (b) describes the NDCG and MMF changes *w.r.t.* accuracy-fairness trade-off co-efficient $\lambda$. Sub-figure (c) conducts the case study on the advantage group News and worst-off group Sports. We show their weight $s_g$, group score $w_g$, and t-SNE embeddings of UNI and FairDual.

**Influence on co-efficient $\lambda$.** Then, we will investigate the impacts of trade-off co-efficient $\lambda$. Figure 3 (b) illustrates that the fairness degree (MMF) increases proportionally with the rise in $\lambda$, aligning with our expectations. However, we also observe that the accuracy increases as $\lambda$ changes from 0 to 5 and then decreases. This phenomenon occurs due to the presence of popularity bias in recommendation datasets (Jiang et al., 2024). A relatively higher fairness degree helps mitigate this bias, leading to increased accuracy. However, when $\lambda$ becomes too large, it inevitably enlarges the Jensen gap, which hurts the model's performance. We also conduct an experiment to analyze the effect of the popularity bias in Appendix N.

**Case study.** Finally, we conduct a case study on the head group *News*, which consistently exhibited superior exposure compared to other groups, in contrast to the tail group *Sports*, which typically had lower exposure levels. Firstly, from the two figures at the top of Figure 3 (c), we observe that as the training progresses, the tail group Sports gradually gains more weight ($s_g$), while the head group News consistently receives relatively low weight. Consequently, this leads to the group scores $w$ of the two groups being close to each other.

At the same time, we visualize the item embeddings using T-SNE (Van der Maaten and Hinton, 2008) for both the baseline UNI and our model FairDual, as shown in the bottom two sub-figures of Figure 3 (c). From the figure, we compute the embedding KL divergence of two different groups between UNI (0.113) and our method FairDual (0.083). This shows that UNI establishes a clear classification line to distinguish between different groups. However, FairDual tends to bring the embeddings of the tail group closer to those of the head group, ultimately increasing the fairness.

**Other experimental analysis.** For analysis of other parameters dual learning rate $\eta$, updating gap $\beta$, user history length $H$, the sample size $Q$, and the impact of the hidden layer numbers, please see the Appendix K. For the computational and storage costs analysis can be seen in Appendix L. We also test the performance of other fairness metric in Appendix M.

## 7 CONCLUSION

In this paper, we theoretically demonstrate that adapting mini-batch training with the objective constrained by group MMF inevitably leads to the Jensen gap, thereby impairing the performance of the RS model. We theoretically and empirically analyze the origins of the Jensen gap by demonstrating that the integration of the MMF constraint disrupts the assumption of sample independence during optimization, leading to a deviation of the loss function from linear additivity. Then, to efficiently bridge the Jensen gap, we develop a large-scale friendly algorithm named FairDual, which employs dual-optimization techniques to minimize the Jensen gap at a sub-linear rate. Extensive experiments conducted on three large-scale recommendation system backbone models using two publicly available datasets show that FairDual consistently outperforms all baseline methods.

ACKNOWLEDGMENTS

This work was funded by the National Key R&D Program of China (2023YFA1008704), the National Natural Science Foundation of China (62472426, 62276248), and the Youth Innovation Promotion Association CAS under Grants No. 2023111. This work was also funded for building world-class universities (disciplines) of Renmin University of China. Supported by the Outstanding Innovative Talents Cultivation Funded Programs 2025 of Renmin University of China.

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

APPENDIX

## A  PROOF OF THEOREM 1

*Proof.* Let $\boldsymbol{A} \in \mathbb{R}^{|\mathcal{I}| \times |\mathcal{G}|}$ is the item-group adjacent matrix, and $\boldsymbol{A}_{ig} = 1$ indicates item $i \in \mathcal{I}_g$, and 0 otherwise. Let $\boldsymbol{w} \in \mathbb{R}^{|\mathcal{I}|} = [-\sum_{u \in \mathcal{U}} c_{u,i} \log(\hat{c}_{u,i})]_{i \in \mathcal{I}}$.

Firstly, if an item belongs to multiple groups, it often has a greater impact on other items in the model. Therefore, we will conduct row normalization on the adjacency matrix $A$ with size $I \times G$ to mitigate this influence: we conduct $\hat{A}$ is the row-normalized matrix for $\boldsymbol{A}$: $\hat{A} = \text{diag}(\boldsymbol{A}\mathbf{1})^{-1}\boldsymbol{A}$. $\text{diag}(\boldsymbol{x})$ denotes to construct a diagonal matrix based on vector $\boldsymbol{x}$.

Then, in RS, since the ranking list $L_K(u)$ is selected according to the highest preference score $c_{u,i}$, therefore we can re-write Equation (1) as:

$$
\begin{aligned}
\min \quad & \mathbf{1}^\top(\hat{\boldsymbol{A}}^\top \boldsymbol{w}) \\
\text{s.t.} \quad & s_g = \sum_{u \in \mathcal{U}} \sum_{i \in L_K(u)} -\frac{\mathbb{I}(i \in \mathcal{I}_g)}{n_i} c_{u,i} \log(\hat{c}_{u,i}) \le m_g M, \forall g \in \mathcal{G} \\
& \sum_{i \in \mathcal{I}} c_{u,i} \le K, \forall u \in \mathcal{U}.
\end{aligned}
\tag{11}
$$

Then we can still write the Equation as

$$
\begin{aligned}
\min \quad & \mathbf{1}^\top(\hat{\boldsymbol{A}}^\top \boldsymbol{w}) \\
\text{s.t.} \quad & \max_{g \in \mathcal{G}}(\hat{\boldsymbol{A}}^\top \boldsymbol{w})_g \le m_g M.
\end{aligned}
\tag{12}
$$

Due to Lagrange dual method (Boţ et al., 2008), we can still convert the problem as:

$$
\min \max_\lambda -\mathbf{1}^\top(\hat{\boldsymbol{A}}^\top \boldsymbol{w}) + \lambda(\max_{g \in \mathcal{G}} \frac{(\hat{\boldsymbol{A}}^\top \boldsymbol{w})_g}{m_g} - M).
\tag{13}
$$

Therefore, we can always find a $\lambda \ge 0$ ($\lambda$ value relates to the value of $M$), such that we can directly optimize

$$
\min \mathbf{1}^\top \hat{\boldsymbol{A}}^\top \boldsymbol{w} + \lambda \max_{g \in \mathcal{G}} \frac{(\hat{\boldsymbol{A}}^\top \boldsymbol{w})_g}{m_g}.
\tag{14}
$$

Let $\boldsymbol{\gamma} \in \mathbb{R}^{|\mathcal{G}|}$ be the vector $[1/m_1, 1/m_2, \cdots, 1/m_{|\mathcal{G}|}]$, then the equation can be written as:

$$
\min \mathbf{1}^\top \hat{\boldsymbol{A}}^\top \boldsymbol{w} + \lambda \max_{g \in \mathcal{G}} \boldsymbol{\gamma}_g(\hat{\boldsymbol{A}}^\top \boldsymbol{w})_g.
\tag{15}
$$

Furthermore, the minimum function $\max(\cdot)$ can be viewed as the infinite norm function:

$$
\max \boldsymbol{x} = \lim_{t \to \infty} (\mathbf{1}^\top \boldsymbol{x}^t)^{1/t}.
$$

Then we consider the following function:

$$
g(\boldsymbol{x}; \boldsymbol{k}; s) = (\boldsymbol{k}^\top \boldsymbol{x}^{1+s})^{\frac{1}{1+s}},
$$

where $\mathbf{0} \le \boldsymbol{x} \le M\mathbf{1}$. Therefore, Equation (15) can be regarded as a linear trade-off between two points with the $\lambda \ge 0$ as the trade-off coefficient:

$$
\min_{\boldsymbol{w} \in \mathcal{W}} g(\hat{\boldsymbol{A}}^\top \boldsymbol{w}; \mathbf{1}; 0) + \lambda g(\hat{\boldsymbol{A}}^\top \boldsymbol{w}; \boldsymbol{\gamma}; \infty).
$$

Since $g(\boldsymbol{x}; t)$ is continuous *w.r.t.* $t$ and the feasible region of $\boldsymbol{x}$ is convex and continuous (because $\boldsymbol{w} \in \mathcal{W}$ is the linear transformation over a simplex space (Lindenstrauss et al., 1978)), there exists a constant number $t \ge 0$ and $\boldsymbol{b} \in \mathbb{R}^{|\mathcal{G}|}$, s.t. Equation (1) can be optimized as:

$$
\mathcal{L} = \min \boldsymbol{b}^\top(\hat{\boldsymbol{A}}^\top \boldsymbol{w})^{1+t}.
$$

This is because $t$ is a constant number and $\boldsymbol{k}$ for optimizing $g(\boldsymbol{x}; \boldsymbol{k}; s)$ is linear w.r.t. to $x^{(1+s)}$ (Since the $x$ is the variable and $s$ is constant).

Note that the specific value of $t$ is an implicit function and cannot be solved explicitly in closed form. This is because according to the fact that the function $g$ is continuous with respect to $s$ over its entire domain and based on the intermediate value theorem for continuous functions, there must exist a $t$ such that the linear combination of the linear functions at the two endpoints equals.

Nonetheless, we emphasize that the subsequent methods and proof strategies are independent of the explicit solution for $t$. As long as there exists a $t \neq 0$, the Jansen gap exists, and as $\lambda$ increases, $t$ will also increase.

$\square$

## B   LEMMA 1

**Lemma 1.** *When $t \geq 0$, let*

$$f(x) = x^{t+1}$$

*where $x > 0$. And*

$$e(i) = \sum_{l=1}^{i} f(y_l)$$

$$\text{s.t.} \quad \sum_{l=1}^{i} y_l \leq c, \quad y_l \geq 0$$

(16)

*where $c$ is a constant number. Then we have when $j \geq i$: we have $\min_{y_j} e(j) \leq \min_{y_i} e(i)$.*

*Proof.* According to the Lagrange dual method, we have

$$\min e(i) = \min \max_{\lambda \geq 0} \sum_{l=1}^{i} f(y_l) + \lambda(\sum_{l=1}^{i} y_l) - \lambda c,$$

then according to the condition of the first derivative equaling zero, we have

$$\frac{\partial e(i)}{\partial y_l} = y_i^t + \lambda = 0, \quad \frac{\partial e(i)}{\partial \lambda} = \sum_{l=1}^{i} y_l - c = 0.$$

Taking these two condition together, we have:

$$y_k = y_m = \frac{c}{i}, \quad \forall k, m = [1, 2, \cdots, i].$$

Therefore, we have

$$\min_{y_j} e(j) - \min_{y_i} e(i) = \min_{y_j} \sum_j f(y_j) - \min_{y_i} \sum_i f(y_i)$$

$$= (\frac{c}{j})^{1+t} - (\frac{c}{i})^{1+t}$$

Then we can see function $\frac{1}{x^{1+t}}$ is a decreasing function function, therefore, $\min_{y_j} e(j) \leq \min_{y_i} e(i)$.

$\square$

## C  PROOF OF THEOREM 2

*Proof.* Under mini-batch sample strategies, we partition the user set $\mathcal{U}$ into $|\mathcal{U}|/B$ subsets and perform optimization on each subset. For each batch, the optimization becomes

$$\mathcal{L}^B = \min \sum_{j=1}^{|\mathcal{U}|/B} \boldsymbol{b}^\top (\hat{\boldsymbol{A}}^\top \boldsymbol{w}_j)^{1+t} \tag{17}$$

$$\text{s.t.} \quad \boldsymbol{w}_{j,i} = -\sum_{u \in \mathcal{U}_j} c_{u,i} \log(\hat{c}_{u,i}), \forall i \in \mathcal{I}, j \in [1, 2, \cdots, |\mathcal{U}|/B],$$

where $\mathcal{U}_b$ is the $b-$th partition of the user set $\mathcal{U}$.

Since the function $f(x) = x^{1+t}$ is not a linear function, we have

$$\sum_{j=1}^{|\mathcal{U}|/B} \boldsymbol{b}^\top (\boldsymbol{A}^\top \boldsymbol{w}_j)^{1+t} \neq \boldsymbol{b}^\top (\boldsymbol{A}^\top \boldsymbol{w})^{1+t}.$$

and we can get the Jensen gap

$$J(B) = |\mathcal{L}^B - \mathcal{L}| \neq 0.$$

Then we will observe how $e(B)$ changes *w.r.t.* the mini-batch size $B$.

Let $\boldsymbol{e} = \hat{\boldsymbol{A}}^\top \boldsymbol{w}$, where each element $\boldsymbol{e}_g$ represents the utility (sum of user-item scores) of group $g$. According to the recommendation constraint, we have $\boldsymbol{e}_g \leq L$, meaning that the utility of group $g$ is at least as high as when all items belonging to group $g$ are recommended to the users.

Therefore, taking $f(\boldsymbol{e}_g) = \boldsymbol{e}_g^{1+t}$ into Lemma 1, without loss of generality, when batch size $B_2 \leq B_1$, we have: $|\mathcal{U}|/B_2 \geq |\mathcal{U}|/B_1$, we can easily have:

$$\min \sum_{j=1}^{|\mathcal{U}|/B_2} (\boldsymbol{A}^\top \boldsymbol{w}_j)_g^{1+t} \leq \min \sum_{j=1}^{|\mathcal{U}|/B_1} (\boldsymbol{A}^\top \boldsymbol{w}_j)_g^{1+t} \leq \min \boldsymbol{e}_g^{1+t}.$$

Therefore, we have

$$\min \sum_{j=1}^{|\mathcal{U}|/B_2} \boldsymbol{b}^\top (\boldsymbol{A}^\top \boldsymbol{w}_j)^{1+t} \leq \min \sum_{j=1}^{|\mathcal{U}|/B_1} \boldsymbol{b}^\top (\boldsymbol{A}^\top \boldsymbol{w}_j)^{1+t} \leq \min \boldsymbol{b}^\top \boldsymbol{e}^{1+t}.$$

In other words, the mini-batch size becomes smaller, and we are more likely to underestimate the original loss function that trades off MMF and recommendation accuracy. The recommendation loss underestimation will result in the Jensen gap when optimizing the loss function constraint with MMF.

$\square$

## D  LEMMA 2

**Lemma 2.** *Considering the following function, for $\lambda > 0, L > 0$ and for the d-th dimension variable $\boldsymbol{\mu} \in \mathbb{R}^d$, when $\boldsymbol{\mu} \in \mathcal{M}$:*

$$r(\boldsymbol{\mu}) = \max_{\boldsymbol{x} \leq \boldsymbol{m}} (\min \boldsymbol{x}/\boldsymbol{m} + \boldsymbol{\mu}^\top \boldsymbol{x}/\lambda), \tag{18}$$

*where*

$$\mathcal{M} = \left\{ \boldsymbol{\mu} \left| \sum_{i \in [d]} \boldsymbol{\mu}_i m_i \geq -\lambda, \forall [d] \in \mathcal{S} \right. \right\},$$

*where $\mathcal{S}$ is power set of $[1, 2, \cdots, d]$, i.e., the set of all subsets of $[1, 2, \cdots, d]$.*

*When $\boldsymbol{\mu} \in \mathcal{M}$, the optimization function $r(\cdot)$ has a closed form: $r(\boldsymbol{\mu}) = \boldsymbol{m}^\top \boldsymbol{\mu}/\lambda + 1$, and $\boldsymbol{m} = \arg\max_{\boldsymbol{x} \leq \boldsymbol{m}} (\min \boldsymbol{x}/\boldsymbol{m} + \boldsymbol{\mu}^\top \boldsymbol{x}/\lambda)$.*

*When $\boldsymbol{\mu} \notin \mathcal{M}$, the function $r(\cdot)$ will diverge to $\infty$.*

*Proof.* Let the variable $\boldsymbol{z} = \boldsymbol{x}/\boldsymbol{m} - \boldsymbol{1}$. Then we have:

$$r(\boldsymbol{\mu}) = \max_{\boldsymbol{x} \leq \boldsymbol{m}} \left[ \min \boldsymbol{x}/\boldsymbol{m} + \boldsymbol{\mu}^\top \boldsymbol{x}/\lambda \right]$$

$$= \boldsymbol{\mu}^\top \boldsymbol{m}/\lambda + 1 + \max_{\boldsymbol{z} \leq \boldsymbol{0}} \left[ \min_i \boldsymbol{z}_i + (1/\lambda)\boldsymbol{\mu}^\top (\boldsymbol{z} \odot \boldsymbol{m}) \right],$$

where $\odot$ is the Hadamard product.

Let

$$\boldsymbol{v} = \boldsymbol{m} \odot \boldsymbol{\mu}/\lambda,$$

then we define

$$s(\boldsymbol{v}) = \max_{\boldsymbol{z} \leq \boldsymbol{0}} \left( \min_i \boldsymbol{z}_i + \boldsymbol{z}^\top \boldsymbol{v} \right).$$

From the definition of the region $\mathcal{M}$, we can re-wright $\mathcal{M}$ as

$$\mathcal{M} = \{\boldsymbol{v}| \sum_{i \in [d]} \boldsymbol{v}_i \geq -1, \forall [d] \in \mathcal{S}\}.$$

Suppose that there exists a subset $\mathcal{S} \in [1, 2, \cdots, d]$ such that $\sum_{i \in \mathcal{S}} \mathbf{v}_i < -1$. For any $\epsilon/|\mathcal{S}| > 1$, we can get a feasible solution:

$$\boldsymbol{v}_i = \begin{cases} -\epsilon/|\mathcal{S}|, & i \in \mathcal{S} \\ 0, & otherwise. \end{cases}$$

Then, because such solution is feasible and $\min_i \boldsymbol{z}_i = -\epsilon$, and $|\mathcal{S}| \geq 1$, we obtain that

$$s(\boldsymbol{v}) \geq \min_i \boldsymbol{z}_i - (\epsilon/|\mathcal{S}|)(\sum_{i \in \mathcal{S}} \boldsymbol{v}_i) = -\epsilon(\sum_{i \in \mathcal{S}} \boldsymbol{v}_i + 1/|\mathcal{S}|)$$

$$\geq \epsilon(\sum_{i \in \mathcal{S}} \boldsymbol{v}_i + 1).$$

Let $\epsilon \to \infty$, we have $s(\boldsymbol{v}) \to \infty$.

Then we show that $s(\boldsymbol{\mu}) = 0$ for $\boldsymbol{v} \in \mathcal{M}$. Note that $\boldsymbol{z} = 0$ is feasible. Therefore, we have

$$\min_i \boldsymbol{v}_i \geq s(0) = 0.$$

Then we have $\boldsymbol{z} \leq 0$ and without loss of generality, that the vector $\boldsymbol{z}$ is sorted in increasing order, i.e., $\boldsymbol{z}_1 \leq \boldsymbol{z}_2, \cdots, \leq \boldsymbol{z}_d$. The objective value is

$$s(\boldsymbol{v}) = \boldsymbol{z}_1 + \boldsymbol{v}^\top \boldsymbol{z}$$

$$= \sum_{j=1}^{d} (\boldsymbol{z}_j - \boldsymbol{z}_{j+1}) \left( 1 + \sum_{i=1}^{j} \boldsymbol{v}_j \right) \leq 0.$$

Thus we can have $s(\boldsymbol{\mu}) = 0$ for $\mathbf{v} \in \mathcal{M}$. Finally, we can have $\arg\max_{\boldsymbol{x} \leq \boldsymbol{m}} (\min_g \boldsymbol{x}_g/\boldsymbol{m}_g + \boldsymbol{\mu}^\top \boldsymbol{x}/\lambda) = \boldsymbol{m}$.

$\square$

# E  LEMMA 3

**Lemma 3.** *The feasible space $\mathcal{M}$ of dual variable $\boldsymbol{\mu}$ is convex.*

*Proof.* Suppose $\boldsymbol{\mu} \in \mathcal{M}$, from Lemma 2, we have

$$r(\boldsymbol{\mu}) = \max_{\boldsymbol{x} \leq \boldsymbol{\gamma}} (\min \boldsymbol{x}/\boldsymbol{\gamma} + \boldsymbol{\mu}^\top \boldsymbol{x}/\lambda) < \infty,$$

therefore, for any $\boldsymbol{b} \in \mathbb{R}_+^{|G|}$ and $c > 0$, we have

$$r(\boldsymbol{\mu} + c\boldsymbol{b}) = \max_{\boldsymbol{x} \leq \boldsymbol{\gamma}} (\min \boldsymbol{x}/\boldsymbol{\gamma} + (\boldsymbol{\mu} + c\boldsymbol{b})^\top \boldsymbol{x}/\lambda)$$

$$= r(\boldsymbol{\mu}) + Lc\boldsymbol{b}^\top \boldsymbol{1} < \infty.$$

Therefore, $\boldsymbol{\mu} + c\boldsymbol{b} \in \mathcal{M}$.

$\square$

# F  PROOF OF THEOREM 3

*Proof.* Let $\boldsymbol{e} = \boldsymbol{A}\boldsymbol{w}$, where each element $\boldsymbol{e}_g$ measures the ranking score accumulated among group $g$. Let $L$ be the maximum ranking score for each group, *i.e.*, $\boldsymbol{e} \leq L\boldsymbol{1}$.

According to the proof in Theorem 1, we can see the Equation (1) can be written as:

$$\min \sum_{u \in \mathcal{U}} \sum_{i \in \mathcal{I}} c_{u,i} \log(\hat{c}_{u,i}) + \lambda(\max_{g \in \mathcal{G}} \boldsymbol{e}_g/m_g)$$

$$\text{s.t.} \quad \boldsymbol{e}_g = -\frac{\mathbb{I}(i \in \mathcal{I}_g)}{n_i} c_{u,i} \log(\hat{c}_{u,i}), \forall g \in \mathcal{G}.$$

Then the equation can be re-written as:

$$\max_{\hat{c}_{u,i}} \sum_{u \in \mathcal{U}} \sum_{i \in \mathcal{I}} c_{u,i} \log(\hat{c}_{u,i}) + \lambda(\min_{g \in \mathcal{G}} \boldsymbol{e}_g/m_g)$$

$$\text{s.t.} \quad \boldsymbol{e}_g = \sum_{u \in \mathcal{U}} \sum_{i \in \mathcal{I}} \mathbb{I}(i \in \mathcal{I}_g) c_{u,i} \log(\hat{c}_{u,i}), \forall g \in \mathcal{G}. \tag{19}$$

Then e can utilize the Lagrangian condition (Balseiro et al., 2021) to decompose the relation between $\boldsymbol{e}$ and model prediction $\hat{c}_{u,i}$ in Equation (19):

$$\max_{\hat{c}_{u,i}} \min_{\boldsymbol{\mu}} \sum_{u \in \mathcal{U}} \sum_{i \in \mathcal{I}} c_{u,i} \log(\hat{c}_{u,i}) + \lambda(\min_{g \in \mathcal{G}} \boldsymbol{e}_g/m_g) - \sum_{g \in \mathcal{G}} \boldsymbol{\mu}_g \left( \boldsymbol{e}_g - \sum_{u \in \mathcal{U}} \sum_{i \in \mathcal{I}} \mathbb{I}(i \in \mathcal{I}_g) c_{u,i} \log(\hat{c}_{u,i}) \right)$$

$$\leq \min_{\boldsymbol{\mu}} \max_{\hat{c}_{u,i}} \sum_{u \in \mathcal{U}} \sum_{i \in \mathcal{I}} c_{u,i} \log(\hat{c}_{u,i}) + \lambda(\min_{g \in \mathcal{G}} \boldsymbol{e}_g/m_g) + \sum_{g \in \mathcal{G}} \boldsymbol{\mu}_g \left( \boldsymbol{e}_g - \sum_{u \in \mathcal{U}} \sum_{i \in \mathcal{I}} \mathbb{I}(i \in \mathcal{I}_g) c_{u,i} \log(\hat{c}_{u,i}) \right)$$

$$= \min_{\boldsymbol{\mu}} \max_{\hat{c}_{u,i}} \left( \sum_{u \in \mathcal{U}} \sum_{i \in \mathcal{I}} (1 - \sum_{g \in \mathcal{G}} \boldsymbol{\mu}_g \mathbb{I}(i \in \mathcal{I}_g)) c_{u,i} \log(\hat{c}_{u,i}) \right) + \lambda \min_g \boldsymbol{e}_g/m_g + \boldsymbol{\mu}^\top \boldsymbol{e}$$

$$= \min_{\boldsymbol{\mu}} \max_{\hat{c}_{u,i}} \left( \sum_{u \in \mathcal{U}} \sum_{g \in \mathcal{G}} (1 - \boldsymbol{\mu}_g) \sum_{i \in \mathcal{I}_g} c_{u,i} \log(\hat{c}_{u,i}) \right) + \lambda \min_g \boldsymbol{e}_g/m_g + \boldsymbol{\mu}^\top \boldsymbol{e}. \tag{20}$$

From the Equation (20), we can observe that the recommendation task constrained by max-min fairness can be viewed as a re-weighting approach across different groups on the original loss function solely optimized for accuracy:

$$\mathcal{L} = \min -\sum_{u \in \mathcal{U}} \sum_{g \in \mathcal{G}} \boldsymbol{s}_g \sum_{i \in \mathcal{I}_g} c_{u,i} \log(\hat{c}_{u,i}),$$

where the fairness weight $\boldsymbol{\mu}$ is determined by

$$\boldsymbol{\mu} = \arg\min_{\boldsymbol{\mu} \in \mathcal{M}} \left( \max \sum_{u \in \mathcal{U}} \sum_{g \in \mathcal{G}} \boldsymbol{s}_g \sum_{i \in \mathcal{I}_g} c_{u,i} \log(\hat{c}_{u,i}) + \lambda r^*(\boldsymbol{\mu}) \right),$$

$$r^*(\boldsymbol{\mu}) = \max_{\boldsymbol{w} \leq \boldsymbol{m}} \left( \min_g (\boldsymbol{A}\boldsymbol{w})_g m_g + \boldsymbol{A}^\top \boldsymbol{w} \boldsymbol{\mu}/\lambda \right) = \boldsymbol{m}^\top \boldsymbol{\mu}/\lambda + 1.$$

To make sure the functions do not diverge, we need to ensure $r^*(\boldsymbol{\mu}) < \infty$. Taking the $r^*(\boldsymbol{\mu})$ into Lemma 2, we show $\boldsymbol{\mu} \in \mathcal{M}$, where

$$\mathcal{M} = \left\{ \boldsymbol{\mu} \, \middle| \, \sum_{g \in \mathcal{S}} \boldsymbol{\mu}_g m_g \geq -\lambda, \forall \mathcal{S} \in \mathcal{G}_s \right\},$$

where $\mathcal{G}_s$ is power set of $\mathcal{G}$, i.e., the set of all subsets of $\mathcal{G}$.

$\square$

## G  PROOF OF THEOREM 4

*Proof.* **We first bound the performance on the primal space.**

Let $N = \frac{|\mathcal{U}|}{B}$ be the total batch number. Considering the $j-$th batch, we have the accuracy loss function without fairness at $j-$th batch as:

$$\mathcal{L}^j(\text{ACC}) = (\boldsymbol{s}^j + \boldsymbol{A}^j \boldsymbol{\mu}^j)^\top \boldsymbol{l}^j = \boldsymbol{1}^\top \boldsymbol{l}^j,$$

and the max-min fairness loss function will become:

$$\mathcal{L}^j(\text{Fair}) = r^*(\boldsymbol{\mu}) - (\boldsymbol{\mu}^j)^\top \boldsymbol{e}/\lambda,$$

therefore, the overall loss across $\mathcal{L}^B$ on the primal space utilizing batch training will become:

$$\begin{aligned}
N\mathbb{E}_j[\mathcal{L}^j] &= N\mathbb{E}_j[\mathcal{L}^j(\text{ACC}) + \lambda\mathcal{L}^j(\text{Fair})] \\
&= N\mathbb{E}_j[(\boldsymbol{s}^j + \boldsymbol{A}^j\boldsymbol{\mu}^j)^\top \boldsymbol{l}^j + \lambda r^*(\boldsymbol{\mu}) - (\boldsymbol{\mu}^j)^\top \boldsymbol{e}] \\
&= \mathcal{L}'^B - N\mathbb{E}_j[(\boldsymbol{\mu}^j)^\top (\boldsymbol{e} - (\boldsymbol{A}^j)^\top \boldsymbol{l}^j)].
\end{aligned}$$

The term

$$w(\boldsymbol{\mu}^j) = (\boldsymbol{\mu}^j)^\top (\boldsymbol{e} - (\boldsymbol{A}^j)^\top \boldsymbol{l}^j)$$

is considered as the complementary slackness in dual theory (Churchman et al., 1957), which captures error from the dual transformation. And $\mathcal{L}'^B$ is the same in Equation (5). Therefore, the original loss can be viewed as the dual form augmented with a complementary slackness form.

**Then we utilize the online gradient descent to bound the complementary slackness.**

Let $\boldsymbol{\mu} = \sum_{j=1}^N \boldsymbol{\mu}^j$, then the loss without dividing the full dataset into batches can be represented as:

$$\mathcal{L} = \mathcal{L}' - w(\boldsymbol{\mu}).$$

After observing the dual form of $\mathcal{L}'$, we can see the $\mathcal{L}'$ is linear *w.r.t.* dual variable $\boldsymbol{\mu}$, therefore, we have

$$\mathcal{L}' = \mathcal{L}'^B,$$

and the Jensen gap

$$J(B) = |\sum_{j=1}^N w(\boldsymbol{\mu}^j) - w(\boldsymbol{\mu})|.$$

Given $\|\widetilde{g}^j\|_2 \leq G$, for all $j$, we have:

$$\|\mathbf{g}^j\|_2 = \|(1-\alpha)\sum_{s=1}^j \alpha^{j-s}(\widetilde{\mathbf{g}}^s)\|_2 \leq G.$$

Next, we will bound the value of $G$. Firstly, according to the dual gradient descent, we have:

$$\widetilde{\boldsymbol{g}}^j = \partial(\boldsymbol{s}^j\mathcal{L}^j + \lambda r^*(\boldsymbol{\mu}^j)) = -(\mathbf{A}^j)^\top \widetilde{\boldsymbol{w}} + \boldsymbol{\gamma}_j.$$

where $\boldsymbol{\gamma}_j$ is the remain maximum loss column at $j-$th updating batch.

Therefore, we have the each element of $\widetilde{\boldsymbol{w}}_b$ has the bound of

$$\widetilde{\boldsymbol{w}}_b \leq K,$$

since each user can obtain a maximum ranking score of 1 for each preferred item in the ranking list with a size of $K$. Typically, the group size is smaller than batch size ($|\mathcal{G}| < B$) and there exists $c = \max_g m_g$ (typically, $m_g$ is proportional to the group size $|\mathcal{G}|$. Then we get

$$\|\widetilde{\boldsymbol{g}}^j\|_2^2 \leq |\mathcal{G}|(c + K)^2 \leq L|\mathcal{G}|^2,$$

where $L > 0$.

According to the Theorem 2 in Balseiro et al. (2021), we have

$$J(B) = |\sum_{j=1}^{N} w(\boldsymbol{\mu}^j) - w_t(\boldsymbol{\mu})| \leq \frac{H}{\eta} + \frac{G^2}{(1-\alpha)\sigma}\eta\frac{|\mathcal{U}|}{B} + \frac{G^2}{2(1-\alpha)^2\sigma\eta}$$

$$= \frac{H}{\eta} + \frac{|\mathcal{U}|L|\mathcal{G}|^2}{B(1-\alpha)\sigma}\eta + \frac{L|\mathcal{G}|^2}{2(1-\alpha)^2\sigma\eta}$$

where function $\|\cdot\|_2^2$ is $\sigma-$strongly convex. When setting learning rate $\eta = O(B^{-1/2})$, the Jensen Bound is comparable with $O(B^{-1/2})$.

$\square$

## H   GENERALIZABILITY TO OTHER FORMS OF FAIRNESS

In fact, our method can be easily generalized to the user group level by replacing the adjacency matrix with a user-side equivalent while keeping the rest unchanged. For the two-sided form, it simply requires introducing two coefficients, $\lambda_1$ and $\lambda_2$, and applying two independent dual gradient descent updates as described in our algorithm.

In Theorem 1, we demonstrate that our optimization objective is equivalent to the power-family fairness framework, which encompasses mainstream fairness definitions such as Entropy Fairness, $\alpha$-Fairness, and Theil Index Lan and Chiang (2011). Consequently, our method is highly adaptable and can be generalized to various fairness objectives within this framework.

## I   DETAILS OF EXPERIMENTAL SETTINGS

Here we will provide the details of experimental settings.

**Detailed Implementation Details.**

- Environment: our experiments were implemented using Python 3.9 and PyTorch 2.0.1+cu117 (Paszke et al., 2017). All experiments were conducted on a server with an NVIDIA A5000 running Ubuntu 18.04. We implement FairDual with the cvxpy (Diamond and Boyd, 2016) for optimization.
- Hyper-parameter settings: the learning rate $\eta \in [1e^{-2}, 1e^{-4}]$ (results shown in Figure 5), and trade-off factor $\lambda \in [0, 10]$ (results shown in Figure 3). We set the $m_g$ as the group size $m_g = |\mathcal{I}_g|$. We also tune sample number $Q \in [50, 400]$ (results shown in the Table 5), historical length $H \in [3, 7]$ (results shown in Table 4), freeze parameter updating gap $\beta \in [128, 3840]$ (results shown in Figure 4).
- LLMs settings: To mitigate the impact of randomness, we set the temperature coefficient to 0.2 for the LLM and ran each model three times, taking the average of the results. Other LLMs settings are: the penalty for frequency is 0.0, and the penalty for presence is 0.0, the maximum generated token number to 1024.
- Used toolkit: For the Non-LLMs-RS backbones, we mainly reference the RecBole toolkit[3]. For the LLMs tuning, we reference the BigRec pipelines [4]. And we have also included our code in the supplementary materials to ensure reproducibility.

**Datasets**. The experiments are conducted on the commonly used two widely used and publicly available recommendation datasets, including:

- MIND (Wu et al., 2020)[5]: it is constructed from user news click behavior logs on the Microsoft News platform. we utilize the major topic category of the news to group the items. The dataset contains 94,057 users, 18,801 items, 124,154 interactions, and 17 groups.

---

[3]https://github.com/RUCAIBox/RecBole

[4]https://github.com/SAI990323/BIGRec

[5]https://microsoftnews.msn.com

- Amazon-Book [6]: The Amazon dataset from the book domain (He and McAuley, 2016) with item grouping based on the "categories" field. As part of the preprocessing (Xu et al., 2024b), groups containing fewer than 50 items are amalgamated into a single group, referred to as the "infrequent group". The dataset contains 15,362,619 users, 1,175,085 items, 1,051,862 interactions, and 25 groups.
- Amazon-Electronic [7]: The Amazon dataset from the electronic products (He and McAuley, 2016) with item grouping based on the "categories" field. As part of the preprocessing (Xu et al., 2024b), groups containing fewer than 50 items are amalgamated into a single group, referred to as the "infrequent group". The dataset contains 728,719 users, 160,052 items, 6,739,590 interactions, and 19 groups.

**Backbones**. For the backbone, we first select three large-scale recommender models:

- **NRMS** (Wu et al., 2019) with 110M parameters utilizes BERT (Devlin et al., 2018) as the feature extractor.
- **RecFormer** (Li et al., 2023) with 150M parameters utilizes LongFormer (Beltagy et al., 2020) to learn text-based representation from items
- **BigRec** (Bao et al., 2023) utilizes Lora techniques (Hu et al., 2021) to fine-tune Llama 2 (Touvron et al., 2023) (with 7B parameters). Note that BigRec only utilizes 1024 samples to train due to large computational cost.

Meanwhile, we also cover three traditional recommender models:

- **BPR** Rendle et al. (2012) utilized a pair-wise loss function to train a matrix factorization model for recommendation.
- **GRU4Rec** Hidasi et al. (2015) utilized gated recurrent unit network to learn the historical behaviors of users.
- **SASRec** Kang and McAuley (2018a) utilized attention network to learn the historical behaviors of users.

**Baselines.**

For the baselines, we choose several fair-aware re-weight baselines that aim to improve group MMF:

- **UNI**: each sample has the same weight during training.
- **DRO** (Hashimoto et al., 2018): every step, the model only optimizes the worst-off groups to enhance grou MMF.
- **S-DRO** (Wen et al., 2022): improves DRO with the distributional shift to optimize group MMF.
- **Prop** (Hu et al., 2023) assigns higher group weight to the samples closer to the decision boundary in each group.
- **IFairLRS** (Jiang et al., 2024) employs the reciprocal of the sum popularity of items within the group as the weight assigned to that group.
- **Maxmin Sample** (Abernethy et al., 2022) applies optimizing techniques to dynamically sample groups.

Meanwhile, we also choose three fair-aware non-LLMs recommender models that aim to improve group fairness:

- **FOCF** (Yao and Huang, 2017) applies a fair-aware regularization loss of different groups into non-LLMs RS.
- **Reg** (Kamishima and Akaho, 2017) penalizes the squared difference between the average scores of two groups for all positive user-item pairs into non-LLMs RS.
- **FairNeg** (Chen et al., 2023a) proposed a negative sampling way for pair-wise recommendation into non-LLMs RS. Note that FairNeg only can be applied to pair-wise RS models.

---

[6] http://jmcauley.ucsd.edu/data/amazon/
[7] http://jmcauley.ucsd.edu/data/amazon/

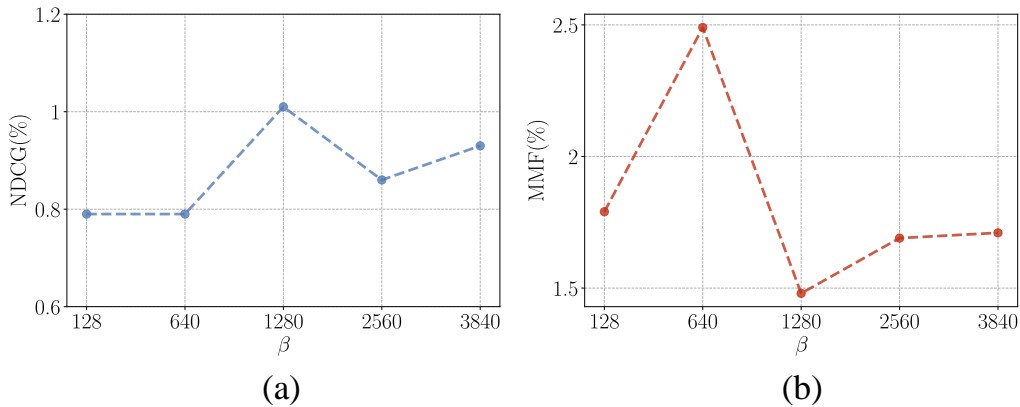

Figure 4: Sub-figure (a) and (b) describe the NDCG and MMF changes *w.r.t.* freeze parameter updating gap $\beta$.

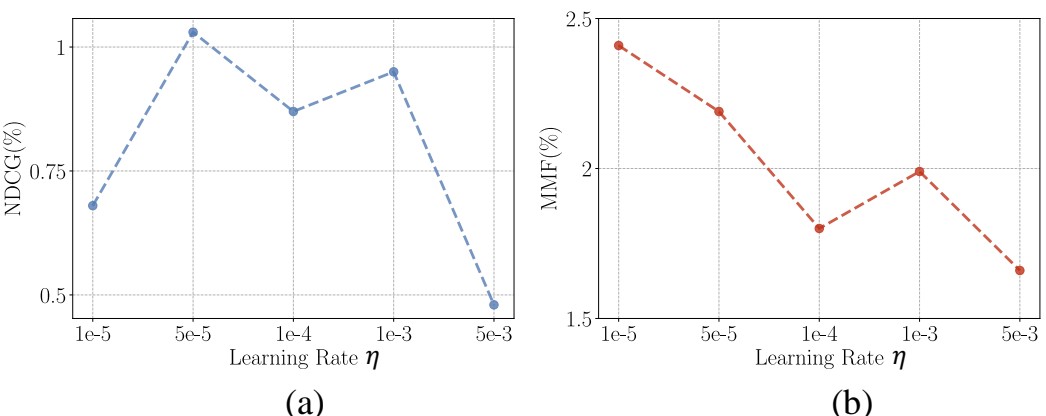

Figure 5: Sub-figure (a) and (b) describe the NDCG and MMF changes *w.r.t.* dual learning rate $\eta$.

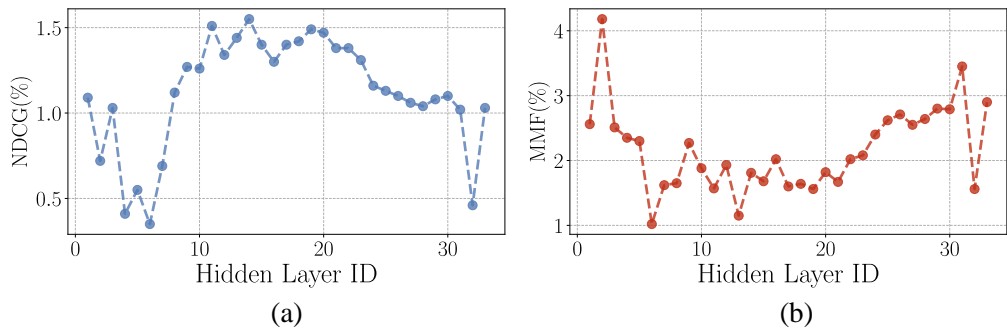

Figure 6: Sub-figure (a) illustrates the density distribution of embeddings for each hidden layer of Llama2. Sub-figures (b) and (c) depict the changes in NDCG and MMF metrics when different hidden layers are utilized to represent user or item embeddings.

Table 3: Performance comparisons between ours under other non-LLMs backbones on MIND dataset. The ∗ means the improvements are statistically significant (t-tests and $p$-value $< 0.05$). The bold number indicates that the accuracy value exceeds that of all the baselines.

| Models/Metrics | | top-5 | | | top-10 | | | top-20 | | |
| --- | --- | --- | --- | --- | --- | --- | --- | --- | --- | --- |
| | | NDCG (%) | MRR (%) | MMF (%) | NDCG (%) | MRR (%) | MMF (%) | NDCG (%) | MRR (%) | MMF (%) |
| **BPR** | DRO | 0.73 | 0.62 | 12.9 | 0.87 | 0.72 | 11.8 | 1.12 | 0.79 | 12.9 |
| | Prop | 0.42 | 0.32 | 0.05 | 0.57 | 0.38 | 0.06 | 0.95 | 0.48 | 10.0 |
| | S-DRO | 0.67 | 0.61 | 3.88 | 0.84 | 0.68 | 6.87 | 1.04 | 0.73 | 12.03 |
| | IFairLRS | 0.68 | 0.57 | 0.13 | 0.77 | 0.61 | 0.23 | 1.07 | 0.69 | 1.38 |
| | Maxmin sample | 0.66 | 0.58 | 6.54 | 0.81 | 0.64 | 8.8 | 1.05 | 0.71 | 10.87 |
| | FOCF | 0.40 | 0.32 | 0.05 | 0.57 | 0.38 | 0.07 | 0.95 | 0.48 | 10.0 |
| | Reg | 0.67 | 0.61 | 3.27 | 0.83 | 0.67 | 5.89 | 1.06 | 0.73 | 11.25 |
| | FairNeg | 0.72 | 0.63 | 6.07 | 0.91 | 0.71 | 8.8 | 1.21 | 0.79 | 12.64 |
| | **Ours** | **0.76**∗ | **0.64**∗ | **11.84**∗ | **0.94**∗ | **0.72** | **13.87**∗ | **1.27**∗ | **0.81** | **14.6**∗ |
| **GRU4Rec** | DRO | 0.56 | 0.56 | 0.86 | 0.76 | 0.64 | 5.56 | 1.13 | 0.71 | 10.7 |
| | Prop | 0.42 | 0.35 | 7.94 | 0.63 | 0.44 | 10.19 | 0.90 | 0.51 | 13.10 |
| | S-DRO | 0.45 | 0.36 | 11.42 | 0.67 | 0.44 | 12.05 | 0.97 | 0.53 | 13.15 |
| | IFairLRS | 0.45 | 0.38 | 7.12 | 0.68 | 0.47 | 9.21 | 1.02 | 0.56 | 11.70 |
| | Maxmin sample | 0.43 | 0.33 | 10.9 | 0.62 | 0.41 | 14.27 | 0.91 | 0.48 | 13.06 |
| | FOCF | 0.56 | 0.41 | 5.62 | 0.79 | 0.63 | 7.11 | 1.10 | 0.70 | 10.29 |
| | Reg | 0.45 | 0.37 | 6.93 | 0.67 | 0.46 | 8.60 | 1.02 | 0.55 | 10.92 |
| | **Ours** | **0.59**∗ | **0.47**∗ | **12.13**∗ | **0.85**∗ | **0.68**∗ | **12.77**∗ | **1.16**∗ | **0.76**∗ | **14.09**∗ |
| **SASRec** | DRO | 0.54 | 0.40 | 8.07 | 0.72 | 0.47 | 11.34 | 1.11 | 0.57 | 12.26 |
| | Prop | 0.54 | 0.45 | 11.69 | 0.80 | 0.55 | 12.10 | 1.16 | 0.57 | 13.01 |
| | S-DRO | 0.49 | 0.40 | 10.66 | 0.74 | 0.49 | 11.64 | 1.09 | 0.59 | 14.02 |
| | IFairLRS | 0.58 | 0.57 | **12.63** | 0.60 | 0.58 | 12.35 | 0.62 | 0.58 | 13.73 |
| | Maxmin sample | 0.56 | 0.47 | 9.05 | 0.74 | 0.54 | 12.45 | 1.09 | 0.64 | 14.06 |
| | FOCF | 0.47 | 0.46 | 10.52 | 0.50 | 0.47 | 12.73 | 0.53 | 0.48 | 14.46 |
| | Reg | 0.47 | 0.38 | 9.42 | 0.70 | 0.47 | 9.52 | 1.03 | 0.55 | 10.91 |
| | **Ours** | **0.64**∗ | **0.63**∗ | 11.98 | **0.78**∗ | **0.64**∗ | **13.08**∗ | **1.31**∗ | **0.67**∗ | **14.51**∗ |

Table 4: We conduct empirical experiments to show the effect of the length of item-clicked sequences. The experiments are conducted on the MIND dataset under BigRec backbones.

| History length $H$ | NDCG@5 (%) | MMF@5 (%) | NDCG@10 (%) | MMF@10 (%) | NDCG@20 (%) | MMF@20 (%) |
| --- | --- | --- | --- | --- | --- | --- |
| 3 | 0.79 | 1.88 | 1.35 | 2.79 | 1.85 | 3.23 |
| 4 | 0.81 | 1.63 | 1.36 | 2.65 | 1.99 | 3.21 |
| 5 | 1.15 | 2.82 | 1.69 | 2.99 | 2.28 | 3.39 |
| 6 | 1.04 | 2.57 | 1.64 | 2.66 | 2.26 | 3.29 |
| 7 | 1.02 | 3.27 | 1.40 | 3.5 | 2.16 | 4.29 |

## J    MAIN EXPERIMENTS ON NON-LLMS BACKBONES.

We choose three non-LLMs recommender models: **BPR** (Rendle et al., 2012), **GRU4Rec** (Hidasi et al., 2015) and **SASRec** (Kang and McAuley, 2018a). And we also compare three group fair-aware on-LLMs recommender models: **FOCF** (Yao and Huang, 2017), **Reg** (Kamishima and Akaho, 2017), and **FairNeg** (Chen et al., 2023a). These models are compared using traditional recommender system backbones, as detailed in Appendix J.

From Table 3, we further observe that FairDual consistently surpasses all baseline methods across various datasets under non-LLM backbones and different top-K ranking sizes. These results demonstrate that FairDual remains highly effective even when applied to non-LLM-based models.

Also note that another widely used loss function is the BPR loss Rendle et al. (2012), which aims to increase the distance between positive and negative samples. Interestingly, from the table, we can observe that our methods can also be applied to this loss, as BPR loss is a convex function with respect to positive items, and our dual formulation remains valid.

## K    ANALYSIS FOR HYPER-PARAMETERS

We also conduct analysis for other important hyper-parameters of FairDual on MIND dataset under BigRec base models.

**Inference on Updating Gap $\beta$.** We first will investigate the impacts of freeze parameter updating gap $\beta$. As shown in Figure 4, we can observe that the accuracy degree (NDCG) increases when $\beta \in [128, 1280]$ and then drops slightly when $\beta \in [1280, 3840]$. Similarly, we can observe that the

Table 5: We conduct empirical experiments to show the effect of the sample size $Q$. The experiments are conducted on the MIND dataset under BigRec backbones.

| sample size $Q$ | 50 | 100 | 200 | 300 | 400 | full (unbiased) |
|---|---|---|---|---|---|---|
| NDCG(%) | 1.08 | 1.08 | 1.15 | 1.19 | 1.19 | 1.29 |
| MMF(%) | 1.2 | 1.28 | 2.18 | 2.10 | 2.29 | 2.31 |

Table 6: The convergence time (performance stabilizing within 50 steps) of our method compared to other baselines under BigRec backbones on the MIND dataset.

| Model | DRO | Prop | S-DRO | IFairLRS | **FairDual(ours)** | Improvment |
|---|---|---|---|---|---|---|
| Convergence time | 10.1h | 11.7h | 7.9h | 7.1h | **5h** | 28.5% |

fairness degree (MMF) increases when $\beta \in [128, 640]$ and then drops with a large margin when $\beta \in [640, 3840]$. The results align with our expectations: excessively frequent updates can lead to instability during training, while infrequent updates may cause the model to miss new ranking patterns, ultimately affecting performance negatively.

**Inference on dual learning rate** $\eta$. We then investigate the impacts of dual learning rate $\eta$. As shown in Figure 5, we can observe that the accuracy degree (NDCG) increases when $\eta \in [1e^{-5}, 5e^{-5}]$ and then drops when $\eta \in [5e^{-5}, 5e^{-3}]$. On the other hand, the fairness performance drops when $\eta$ goes larger. The results demonstrate that the learning rate $\eta$ serves as a trade-off factor: excessively large values detrimentally affect both accuracy and fairness, whereas excessively small values improve fairness at the expense of accuracy in recommendation system models.

**Performances under Different Hidden Layers.** In this experiment, we aim to analyze the FairDual performance under different hidden layers in Llama2. We test the NDCG and MMF performance when we utilize different hidden layers to represent user or item embeddings. From Figure 6 (a) and (b), it is evident that the accuracy (NDCG) and fairness (MMF) trends exhibit distinct patterns: accuracy performance initially ascends, peaking in the middle layer before gradually declining, whereas fairness performance initially descends, hitting a nadir before steadily increasing.

This phenomenon can be interpreted as follows: in the initial layers, which are not yet fully trained, the recommendation system tends to suggest more random items, resulting in lower accuracy but higher fairness. As the layers deepen, the accuracy increases, but it also tends to recommend more unipolar items. Eventually, as the layers approach the last layer, our FairDual model emphasizes fairness more by adjusting the weights for the weaker groups. This will also help us to better understand the mechanisms of FairDual.

**Performances under different lengths** $H$ **of item-clicked sequences.** In Table 4, we conduct the empirical experiments to show the effect for the length of item-clicked sequences. The experiments are conducted on MIND dataset under BigRec backbones.

From the experiments, we can observe that the length of history is a trade-off factor for the methods: initially, increasing the length improves accuracy and fairness, but once it reaches a peak, performance begins to drop. We analyze the reason as follows: the length of history sequences indeed influences performance. Sequences that are too short make it difficult to learn user preferences, while sequences that are too long increase computational costs and risk hitting the prompt limit of LLMs.

**Performances under different sample sizes** $Q$. Intuitively, a larger $Q$ provides a more accurate gradient estimation but also incurs higher computational costs. We have conducted experiments to evaluate the impact of $Q$ and will present the results. The results were conducted under the same settings as the analysis section. The experiments were conducted under BigRec on MIND dataset with ranking size $K = 5$.

From the results in Table 5, we observe that increasing the sample value $Q$ leads to improvements in both accuracy and fairness performance. However, in LLM-based recommender systems, a larger $Q$ significantly increases training time (with each item requiring an additional 1.5 seconds) and storage

Table 7: Performances of other fairness metric Gini Index.

| Models | GINI@5 | GINI@10 | GINI@20 |
|---|---|---|---|
| Prop | 0.488 | 0.488 | 0.472 |
| DRO | 0.511 | 0.476 | 0.487 |
| SDRO | 0.503 | 0.478 | 0.453 |
| IFairLRS | 0.458 | 0.454 | 0.448 |
| **FairDual(ours)** | **0.444** | **0.450** | **0.441** |

Table 8: Popularity bias effect utilizing Inverse Propensity Score (IPS)-based Xu et al. (2022) reweighting method.

| $\lambda$ | 0.1 | 1 | 2 | 5 |
|---|---|---|---|---|
| NDCG(%) | | | | |
| IPS | 0.58 | 0.58 | 0.58 | 0.58 |
| FairDual | 0.53 | 0.60 | 0.57 | 0.67 |
| FairDual+IPS | 0.59 | 0.56 | 0.56 | 0.58 |
| MMF(%) | | | | |
| IPS | 12.63 | 12.63 | 12.63 | 12.63 |
| FairDual | 4.50 | 11.98 | 13.46 | 13.76 |
| FairDual+IPS | 10.90 | 12.40 | 12.40 | 14.36 |

space. Different applications should select appropriate Q values based on their specific accuracy, fairness requirements, and computational constraints.

## L    COMPUTATIONAL AND STORAGE COSTS

In Table 6, we measured the convergence time (performance stabilizing within 50 steps) of our method compared to other baselines under BigRec backbones on MIND dataset.

Firstly, we all have parameters of the same magnitude (i.e., group size parameters (hundred level), which are in the range of hundreds and negligible compared to the backbone (million level)). Our method only requires additional space for $Q$ item embeddings and extra training time ($1.5Q$s). Applications can trade off $Q$ based on available resources (as discussed in a previous response).

Secondly, as observed in Table 6 of the original paper, although there is an additional time overhead per round, our convergence speed accelerates by 30% compared to the best baseline. This 30% improvement in convergence speed is highly significant for industrial applications, along with enhanced performance.

## M    PERFORMANCES ON OTHER FAIRNESS METRICS

We test the performances of another fairness metric Gini Index (Do and Usunier, 2022) Compared to the baselines (Table 7) on MIND datasets. Note that a smaller Gini Index means more fairness. From the results, we can observe that our model can still perform well on other fairness metrics. We believe our paper can help other researchers explore its applicability to various loss functions, and other fairness metrics, which is also our contribution to the communities.

## N    EFFECT OF POPULARITY BIAS

Since the popularity bias will influence the accuracy estimation in real dataset shown in Figure 3, we conduct the experiments on a relatively light transformer-based SASRec (Kang and McAuley, 2018a)

backbones and MIND datasets. We apply the Inverse Propensity Score (IPS)-based Xu et al. (2022) to our method to see whether it can improve our methods.

Table 8 shows the results on $K = 5$ results. From the results, we can observe that when the $\lambda$ is small, adding the IPS will increase the accuracy and fairness with a large margin due to the popularity bias. However, when $\lambda$ is large, the FairDual+IPS will not perform very well. This is because IPS will break the convergence condition of FairDual. Therefore, when $\lambda$ is large, it is preferable not to involve IPS. We will include the related experiments and discussion in the Appendix of the revised paper.

