# OpenReview forum: "Bridging Jensen Gap for Max-Min Group Fairness Optimization in Recommendation"
_ICLR.cc/2025/Conference — ICLR 2025 Poster_

### Official Review · Reviewer_1uT1 · 2024-10-31

**Soundness:** 3
**Presentation:** 3
**Contribution:** 2
**Rating:** 6
**Confidence:** 2

**Summary:**

In this paper, the authors address the challenge of integrating a max-min fairness constraint, which introduces a Jensen gap between the model’s convergence point and the optimal point. They first demonstrate that using mini-batch sampling optimization strategies leads to a Jensen gap that increases as the mini-batch size decreases. To bridge this gap, the authors propose an algorithm that reformulates the original optimization problem through a re-weighting approach, leveraging dual-optimization techniques to update the weights of each group. They theoretically prove that their approach achieves a sublinear convergence rate and numerically demonstrate its effectiveness.

**Strengths:**

- The paper is well written and provides interesting insights.
- The authors well motivated the problem in Section 4 where they established the existence of Jensen gap.
- The theoretical results appear sound. The authors also provided extensive numerical evaluations of their method.

**Weaknesses:**

- It'd be helpful if the authors can provide more interpretations for Theorem 4, the main theoretical result and, in particular, comment on their technical contributions. What is the main technical novelty in attaining this theoretical result? A proof sketch could also be helpful.
- Following the point above, in the numerical experiments, there appears to be some non-monotonic variation of the Jensen gap w.r.t. the batch size. I wonder if the authors can comment on why this is the case. Is this consistent with the theoretical results?

**Questions:**

- Could the results extend to alternative fairness constraints beyond max-min fairness?
- What is the computational complexity of the proposed algorithm and how does it compare with the other baselines?

---

> ### Author Response · Authors · 2024-11-21
> **Rebuttal**
>
> Thanks for your hard work and valuable questions.
>
> W1. It'd be helpful if the authors could provide more interpretations for Theorem 4, the main theoretical result and, in particular, comment on their technical contributions. What is the main technical novelty in attaining this theoretical result? A proof sketch could also be helpful.
> * A1: The main technical novelty lies in **transforming the Jansen gap problem into its dual form** and leveraging dual gradient learning techniques to bound the **complementary slackness** and establish the theoretical bounds. Next is the proof sketch:
>
>
> Firstly, we write **the dual form of the Jansen gap**:
>
> $J(B) = |\sum_{j=1}^Nw({\mu}^j)-w({\mu})|$ and $w({\mu}^j)= ({\mu}^j)^{\top}({e}-({A}^j)^{\top}{l}^j)$
>
> Then we utilize the **online gradient descent** to bound the **complementary slackness** of $w({\mu}^j)$:
> The gradient of $w({\mu}^j)$ can be bounded as $\|\widetilde{{g}}^j\|_2^2 \leq L|\mathcal{G}|^2$.
>
> Finally, we put them together will have our conclusion. We will re-organize the proof and state the main technical novelty.
>
>
> W2: Following the point above, in the numerical experiments, there appears to be some non-monotonic variation of the Jensen gap w.r.t. the batch size. I wonder if the authors can comment on why this is the case. Is this consistent with the theoretical results?
> * A2: Thanks for the question.
>
> Firstly, our previous  monotonic variation of the Jensen gap w.r.t. the batch size is based on ideal fairness aware algorithms. However, in real algorithms, some bias (e.g. gradient and online bias) will influence the process, making it not strict monotonic.
>
> Secondly, our theoretical results provide an upper bound on the Jansen gap, and we observe that the Jansen gap in our method exhibits a monotonic trend with respect to batch size and group size. This aligns with the theoretical results, as the theory does not strictly require the gap to be perfectly monotonic.
>
> W3: Could the results extend to alternative fairness constraints beyond max-min fairness?
> * A3: Thanks for the question. In Theorem 1, we demonstrate that our optimization objective is equivalent to the power-family fairness framework, which encompasses mainstream fairness definitions such as Entropy Fairness, $\alpha$-Fairness, and Theil Index [1].  Consequently, our method is __highly adaptable and can be generalized to various fairness objectives within this framework__.
>
>
> We also test the performances of other fairness metric Gini Index Compared to the baselines (Table 1) on MIND datasets. Note smaller Gini Index means more fairness.
> |Models|GINI@5|GINI@10|GINI@20|
> |-------------------|--------|-------|--------|
> |Prop|0.488|0.488|0.472|
> |DRO|0.511|0.476|0.487|
> |SDRO|0.503|0.478|0.453|
> |IFairLRS|0.458|0.454|0.448|
> |**FairDual(ours)**|**0.444**|**0.450**|**0.441**|
>
> From the results, we can observe that our model can still perform good on other fairness metrics.
>
> We believe our paper can help other researchers explore its applicability to various loss functions, and other fairness metrics, which is also our contributions to the communities.
>
>
> [1] Lan, T., & Chiang, M. (2011). An axiomatic theory of fairness in resource allocation. _George Washington University, http://www. seas. gwu. edu/tlan/papers/fairness. pdf, Tech. Rep_.
>
>
> W4: What is the computational complexity of the proposed algorithm and how does it compare with the other baselines?
> * A4: Thanks for the question.
>
> Firstly, we all have parameters of the same magnitude (i.e., group size parameters, which are in the range of hundreds and negligible compared to the backbone). Our method only requires additional space for Q item embeddings and extra training time (Q * 1.5s). Applications can trade off Q based on available resources:
>
> |Q| 50 | 100 | 200 | 300 | 400 | full (unbiased) |
> |--|--|--|--|--|--|--|
> | NDCG (%)| 1.08 | 1.08 | 1.15 | 1.19 | 1.19 | 1.29|
> | MMF (%)| 1.2 | 1.28 |2.18 |2.10 |2.29 |2.31|
>
> Secondly, as mentioned in Table 3 of the original paper, although there is an additional time overhead per round, our convergence speed accelerates by 30% compared to the best baseline. This 30% improvement in convergence speed is highly significant for industrial applications, along with enhanced performance.
>
> We will add the discussion in the revised paper. Thanks for your question again.

---

> > ### Comment · Reviewer_1uT1 · 2024-12-02
> >
> > Thank you for your detailed response. I will keep my positive assessment.

---

### Official Review · Reviewer_EADi · 2024-11-02

**Soundness:** 3
**Presentation:** 4
**Contribution:** 3
**Rating:** 8
**Confidence:** 4

**Summary:**

In this paper, the authors claimed that current group max-min fairness (MMF) methods would introduce a Jensen gap between the model’s
convergence point and the optimal point. The authors analyzed the existence of Jensen gap theorically and emprically. Then, the authors proposed FairDual, a  dual-optimization approach that guaranteed a bound for the Jesen gap. They conducted comprehensive experiments to show the effectiveness and efficiency of FairDual compared to baselines.

**Strengths:**

- The motivation is well established with theorical and emprical analysises of the Jesen gap in Section 4.
- The method is solid with a guaranteed bound for Jesen gap, and the experiment also showed that the proposed method indeed has lower gap than other baselines.
- The authors conducted complete and comprehensive evaluations, including the effictiveness, Jensen gap analysis, case study, training efficiency and parameter analysis.

**Weaknesses:**

- Baselines. The authors mentioned that there are two types of approaches to bridge the Jensen gap,  re-weighting and optimization-based methods. But the authors mostly compared only re-weighting methods in the experiments, while ignoring optimization-based methods they mentioned in the introduction part, such as  Abernethy et al. 2022, Cousins 2022, Demidovich et al., 2023 and Agarwal et al., 2018. I suggest the authors to add state-of-the-art baselines in optimization-based methods mentioned in the paper.

**Questions:**

- Can authors explain is there a reason that in some cases the MRR is not statistically significant as shown in Table 1 and 2? For example, the MRR in the top-5 results of RecFormer and BigRec is not significant and no improvement, while the improvement of NDCG and MMF is significant. Can authors give some insights on this observation?

- In the visualization of Figure 3(c), the differences in the patterns of the two figures are not quite obvious. The classification boundary of FairDual also seems to exist. Could the authors provide some quantitative results to distinguish the different patterns, such as divergence in the two distributions?

---

> ### Author Response · Authors · 2024-11-21
> **Baselines**
>
> W1: I suggest the authors to add state-of-the-art baselines in optimization-based methods mentioned in the paper.
> * A1:
> We involve __four baselines__ covered from __optimizing-based__ methods,  and __group fair-aware__ recommendation methods. Note FOCF, Reg is designed for the non-LLMs RS models and FairNeg is designed for the pair-wise RS models. We conduct experiments on different LLMs and non-LLMs backbones on MIND dataset.
>
> [1] Max-min sample: applies optimizing techniques to dynamically sample groups.
> [2] FOCF: applies a fair-aware regularization loss of different groups into non-LLMs RS.
> [3] Reg: Penalizes the squared difference between the average scores of two groups for all positive user-item pairs into non-LLMs RS.
> [4] FairNeg: A negative sampling way for pair-wise recommendation into non-LLMs RS.
>
>
>
> For MIND dataset
> | Model| NDCG@5 | MRR@5 | MMF@5 | NDCG@10 | MRR@10 | MMF@10 | NDCG@20 | MRR@20 | MMF@20 |
> |------------------|---------|-------|-------|---------|--------|--------|---------|--------|--------|
> | **Maxmin Sample (new)**| 0.98| 0.75| 2.25| 1.49| 4.43| 0.96| 2.19 | 1.15| 3.13|
> | **FairDual(Ours)**| __1.15__| __0.88__| __2.82__| __1.69__| __1.11__| __2.99__| __2.28__| __1.27__| __3.39__|
>
> For Amazon-book dataset
> | Model| NDCG@5 | MRR@5 | MMF@5 | NDCG@10 | MRR@10 | MMF@10 | NDCG@20 | MRR@20 | MMF@20 |
> |------------------|---------|-------|-------|---------|--------|--------|---------|--------|--------|
> | **Maxmin Sample (new)**| 2.49| 2.31| 6.80| 2.72| 4.43| 6.80| 2.97 | 2.74| 7.5|
> | **FairDual(Ours)**| __3.11__| __2.88__| __8.90__| __3.31__| __2.96__| __9.00__| __3.60__| __3.04__| __8.89__|
>
> BPR[5] backbones:
>
> |Models|NDCG@5|MRR@5|MMF@5|NDCG@10|MRR@10|MMF@10|NDCG@20|MRR@20|MMF@20|
> |-------------------|--------|-------|--------|---------|--------|---------|---------|--------|---------|
> |Prop|0.42|0.32|0.05|0.57|0.38|0.06|0.95|0.48|10.0|
> |DRO|0.73|0.62|12.9|0.87|0.72|11.8|1.12|0.79|12.9|
> |SDRO|0.67|0.61|3.88|0.84|0.68|6.87|1.04|0.73|12.03|
> |IFairLRS|0.68|0.57|0.13|0.77|0.61|0.23|1.07|0.69|1.38|
> |**FOCF(new)**|0.4|0.32|0.05|0.57|0.38|0.07|0.95|0.48|10.0|
> |**Max-min sample(new)**|0.66|0.58|6.54|0.81|0.64|8.8|1.05|0.71|10.87|
> |**Reg(new)**|0.67|0.61|3.27|0.83|0.67|5.89|1.06|0.73|11.25|
> |**FairNeg(new)**|0.72|0.63|6.07|0.91|0.71|8.8|1.21|0.79|12.64|
> |__FairDual(Ours)__|__0.76__|__0.64__|__11.84__|__0.94__|__0.72__|__13.87__|__1.27__|__0.81__|__14.6__|
>
> The results of GRU4Rec [6]:
> | Models| NDCG@5 | MRR@5 | MMF@5 | NDCG@10 | MRR@10 | MMF@10 | NDCG@20 | MRR@20 | MMF@20 |
> |-------------------|--------|-------|-------|---------|--------|--------|---------|--------|--------|
> | UNI| 0.39| 0.36| 5.08| 0.55| 0.42| 6.44| 0.83| 0.5| 9.08|
> | Prop| 0.42| 0.35| 7.94| 0.63| 0.44| 10.19| 0.9| 0.51| 13.1|
> | DRO| 0.56| 0.56| 0.86| 0.76| 0.64| 5.56| 1.13| 0.71| 10.7|
> | SDRO| 0.45| 0.36| 11.42 | 0.67| 0.44| 12.05| 0.97| 0.53| 13.15|
> | IFairLRS| 0.45| 0.38 | 7.12  | 0.68| 0.47 | 9.21 | 1.02| 0.56 | 11.70 |
> | **FOCF (new)**| 0.56| 0.41| 5.62| 0.79| 0.63| 7.11| 1.1| 0.7| 10.29|
> | **Maxmin sample(new)**| 0.43| 0.33| 10.9| 0.62| 0.41| 14.27| 0.91| 0.48| 13.06|
> | **Reg(new)**| 0.45| 0.37| 6.93| 0.67| 0.46| 8.6| 1.02| 0.55| 10.92|
> | **FairDual (Ours)**| __0.59__| __0.47__| __12.13__ | __0.85__| __0.68__| __12.77__| __1.16__| __0.76__| __14.09__|
>
> For the results of SASRec [7]:
> | Models| NDCG@5 | MRR@5 | MMF@5 | NDCG@10 | MRR@10 | MMF@10 | NDCG@20 | MRR@20 | MMF@20 |
> |------------------|--------|-------|-------|---------|--------|--------|---------|--------|--------|
> | UNI| 0.59| 0.5| 10.43 | 0.76| 0.56| 11.91| 1.09| 0.65| 12.94|
> | Prop| 0.54| 0.45| 11.69 | 0.8| 0.55| 12.1| 1.16| 0.57| 13.01|
> | DRO| 0.54| 0.4| 8.07| 0.72| 0.47| 11.34| 1.11| 0.57| 12.26|
> | SDRO| 0.49| 0.4| 10.66 | 0.74| 0.49| 11.64| 1.09| 0.59| 14.02|
> | IFairLRS| 0.58| 0.57| __12.63__ | 0.60| 0.58| 12.35| 0.62| 0.58| 13.73|
> | **FOCF(new)**| 0.47| 0.46| 10.52 | 0.5| 0.47| 12.73| 0.53| 0.48| 14.46|
> | **Minmax_SGD(new)**| 0.56| 0.47| 9.05| 0.74| 0.54| 12.45| 1.09| 0.64| 14.06|
> | **Reg(new)**| 0.47| 0.38| 9.42| 0.7| 0.47| 9.52| 1.03| 0.55| 10.91|
> | **FairDual (Ours)**| __0.64__| __0.63__| 11.98 | __0.78__| __0.64__| __13.08__| __1.31__| __0.67__| __14.51__|
>
> From the results, we can also see that our model can outperform all the baselines except MMF@5 in SASRec, showing our effectiveness on non-LLMs RS models.
>
> [1] J. D. Abernethy, et al.. Active sampling for min-max fairness.
> [2] Yao sirui et al. Beyond parity: Fairness objectives for collaborative filtering.
> [3] Toshihiro Kamishima and Shotaro Akaho. 2017. Considerations on recommendation independence for a find-good-items task.
> [4] Chen, Xiao, et al. Fairly Adaptive Negative Sampling for Recommendations.
> [5] Steffen Rendle et al. "BPR: Bayesian Personalized Ranking from Implicit Feedback.".
> [6] Yong Kiam Tan et al. "Improved Recurrent Neural Networks for Session-based Recommendations.".
> [7] Wang-Cheng Kang et al. "Self-Attentive Sequential Recommendation.".

---

> > ### Author Response · Authors · 2024-11-21
> > **Other questions**
> >
> > W2: Can authors explain is there a reason that in some cases the MRR is not statistically significant as shown in Table 1 and 2? Can authors give some insights on this observation?
> > * A2: This is an interesting question. In MRR, it actually cares about the inverse of the position of the __first relevant item__ while NDCG and MMF computes the overall ranking lists.
> > From the results, we observe that the first relevant item remains almost unchanged across methods; however, FairDual excels at positioning the second and subsequent relevant documents (often belonging to smaller groups) higher in the ranking.
> >
> > This indicates that while the first relevant items typically achieve higher scores, FairDual improves the middle-ranked items in terms of both fairness and accuracy. We will add this discussion to the revised paper—thank you for raising such a thoughtful question.
> >
> > W3: In the visualization of Figure 3(c), the differences in the patterns of the two figures are not quite obvious. Could the authors provide some quantitative results to distinguish the different patterns, such as divergence in the two distributions?
> > * A3: Thanks for the suggestion. We test the KL divergence of two different groups under UNI and our method FairDual:
> >
> > | Models| KL divergence|
> > |------------------|--------|
> > | UNI| 0.113|
> > | FairDual | 0.083|
> >
> > From the results, we observe that our method, FairDual, exhibits a smaller KL divergence, indicating that the embeddings learned through our approach bring the embeddings of the tail group closer to those of the head group, thereby enhancing fairness. We will incorporate these results into the revised version.

---

> > > ### Comment · Reviewer_EADi · 2024-11-22
> > > **Thanks**
> > >
> > > Thanks authors for their comprehensive responses. I believe after solving my questions, the paper can be better. Thus I would like to rise my score.

---

> > > > ### Author Response · Authors · 2024-11-22
> > > >
> > > > Thanks for your hard work.  Your valuable suggestions really help us to improve our paper!
> > > >
> > > > Best,
> > > > Authors

---

### Official Review · Reviewer_w1tq · 2024-11-02

**Soundness:** 2
**Presentation:** 3
**Contribution:** 2
**Rating:** 6
**Confidence:** 2

**Summary:**

This paper observes that the MMF-based optimization introduces a Jensen gap, which will become more pronounced when mini-batch size decreases and group size increases. The authors reformulate MMF into a group-weighted optimization, and solve its dual to minimize Jensen gap. Theoretical analysis reveals that the proposed method achieves a sub-linear convergence rate. Experiments are conducted across three recommendation models on two datasets.

**Strengths:**

1.	The paper provides both theoretical analysis and empirical results for motivation derivation and the effectiveness of the proposed method.
2.	The experiments are conducted across large-scale recommendation models, which aligns well with the stated motivation.
3.	The paper is generally well-written, with a clear structure.

**Weaknesses:**

1.	The assumption of convexity is too strong and impractical for large-scale recommendation models.
2.	Why can the reported NDCG exceed 1, which is theoretically impossible? Also, please specify the number of items in the truncated list K.

**Questions:**

Please refer to Weaknesses.

---

> ### Author Response · Authors · 2024-11-21
> **Rebuttal**
>
> W1: The assumption of convexity is too strong and impractical for large-scale recommendation models.
>
> * A1: Sorry for the confusion. We do not assume that all functions are convex; rather, __we only assume that the optimization objective loss function is convex, which allows it to be transformed into its dual form__. Common recommendation objectives, including BCE loss, Entropy loss, and BPR loss, all exhibit convexity. We will clarify and re-state this assumption regarding convexity.
>
> W2: Why can the reported NDCG exceed 1, which is theoretically impossible? Also, please specify the number of items in the truncated list K.
>
> * A2: Sorry for the confusion.
>
> For the first question, in caption (line 400) of Table 1 and Table 2, we state that __all the numbers with “%” omitted__, which means 1.15 in the Table means 1.15%. Since we evaluate the performance with full ranking (after filtering, for MIND, we rank nearly 1000 items and for Amazon dataset, we rank nearly 4000 items), which will make our ndcg and mrr be this kinds of magnitude [1].
>
> For the second question, The truncated $K$ number in the table denotes the top-$K$, for example, top5 means $K=5$. We will modify the table $top5$ into $K=5$. Thanks for your suggestions.
>
> [1] K. Bao, J. Zhang, W. Wang, Y. Zhang, Z. Yang, Y. Luo, F. Feng, X. He, and Q. Tian. A bi-step grounding paradigm for large language models in recommendation systems. arXiv preprint arXiv:2308.08434, 2023a.

---

> > ### Comment · Reviewer_w1tq · 2024-11-27
> >
> > Thank you for the explanation, which addresses my concerns. Due to my limited expertise in this area, I would like to keep my positive rating but with low confidence.

---

### Official Review · Reviewer_SoKV · 2024-11-03

**Soundness:** 3
**Presentation:** 3
**Contribution:** 3
**Rating:** 8
**Confidence:** 4

**Summary:**

This paper analyze the Group max-min fairness (MMF) constrained optimization problem in recommendation. The authors first explain the Group MMF constrained objective as a non-linear polynomial form, indicating that the Jensen gap is non-negligible in mini-batch sampling based optimization (i.e., the gap between the overall loss and the sum of batched loss). To bridge the Jensen gap, this paper propose a dual optimization method called FairDual. Specifically, they rewrite the objective as a group-weighted BCE loss, and utilize the dual mirror gradient descent algorithm to optimize this loss. They further conduct experimental validation of FairDual's effectiveness and provide a detailed analysis of its proposed theoretical advantages.

**Strengths:**

- The Group MMF studied by the authors is significant for  recommendation fairness research.
- The motivation of the paper is clear, the algorithm introduction is straightforward, and the experimental analysis is detailed.
- The paper employs dual optimization to separate MMF and predicted score, resulting in a simple form of group-weighted BCE loss, and uses the dual mirror gradient descent algorithm for optimization, which is somewhat novel.

**Weaknesses:**

- Theoretical proofs contain parts that require clarification, and the writing of the proof needs further improvement.
- The authors' analysis of the proposed group-weighted BCE loss is insufficient.
- In the implementation of FairDual algorithm, sampling-based ranking may introduce bias that is not adequately discussed.
- The authors conducted experiments on only two datasets, which may not be sufficient to demonstrate the algorithm's generalization.

**Questions:**

**Major Concerns:**

**Questions about Theoretical Analysis and Algorithm.** I have the following questions about the theoretical analysis and algorithm in this paper that need clarification:

- In the proof of Theorem 1 (Appendix A), why is Problem (15), i.e., $\min\mathbf{1}^\top\hat{\boldsymbol{A}}^\top\boldsymbol{w}+\lambda\max_{g\in\mathcal{G}}\gamma_g(\hat{\boldsymbol{A}}^\top\boldsymbol{w})_g$ , equivalent to Problem (2), i.e., $\min\boldsymbol{b}^\top(\hat{\boldsymbol{A}}^\top\boldsymbol{w})^{1+t}$ ? Specifically, considering the limit in the function $g(\cdot; \infty)$ , how is the constant $t$ determined? Providing an explicit solution for $t$ and $\boldsymbol{b}$ is important for Theorems 1 and 2.
- The organization of Lemma 2, Lemma 3, and Theorem 3 (Appendix D-F) is somewhat disorganized. I suggest reorganizing these proofs, e.g.:
  - Place Lemma 3 before Lemma 2, as the conclusion of Lemma 2 depends on Lemma 3.
  - Rewrite the proof of Lemma 2 to explain why the conclusion can be derived from $r(\boldsymbol{\mu} + c\boldsymbol{b}) < \infty$ (i.e., Lemma 3).
- The group-weighted form (4) of the Group MMF-constrained objective in Theorem 3 is concise, but its weight $\boldsymbol{s}_g$ is not. The authors should provide some intuitive explanations for the weight $\boldsymbol{s}_g$ to better elucidate the experimental phenomena (Case study in Section 6.3). For instance, under what circumstances is $\boldsymbol{s}_g$ larger, and when is it smaller?

- In the calculation of $\widetilde{w}$, the authors randomly sample $Q$ items and set $\widetilde{\boldsymbol{w}} _ b=\sum _ {k=1}^K(\boldsymbol{E} ^ j\boldsymbol{e} _ {u _ b}) _ {[k]}$ (cf. line 12 in Algorithm 1, and line 358). I am primarily concerned about the bias caused by sampling-based ranking (although it does not affect the fairness bound given in Theorem 4). Can the authors provide a theoretical analysis of this bias? Alternatively, could the authors change the sampling-based ranking to random sampling of $\boldsymbol{E}^j\boldsymbol{e} _ {u _ b}$ , and test the impact of this bias on the convergence rate of Jensen gap?

**Questions about Experiments.** I have the following concerns about the experiments:

- There are only two datasets utilized in the main results (Tables 1 and 2), which is insufficient. The authors might consider adding one or two widely-used datasets, such as Amazon-Electronic, which can be processed using the same method as in Appendix H.
- In Section 6.3 "Experimental Analysis", the authors find that the accuracy first increases then decreases as $\lambda$ increases, and attribute the phenomenon to the popularity bias. Then, is it possible to apply popularity debias method to the proposed algorithm, e.g., Inverse Propensity Score (IPS)-based reweighting method?

**Minor Concerns:**

- Line 324, $\hat c _{u, i} =  -d(\boldsymbol{e} _u, \boldsymbol{e}  _i)$, should there be $\hat c _{u, i} =  d(\boldsymbol{e} _u, \boldsymbol{e} _i)$ ?
- Line 325, the authors should suppose that $\boldsymbol{e}_u$ and $\boldsymbol{e}_i$ are normalized to make sure $d(\boldsymbol{e}_u, \boldsymbol{e}_i) \leq 1$ , which is relied on by the proof of Theorem 4 (cf. Line 1064).
- Line 357, "The $L$ items’ embeddings are denoted as ...", the $L$ should be $Q$ ?
- Line 979, the minus in $-\mathcal{I}$ should be placed at the loss term.

---

> ### Author Response · Authors · 2024-11-21
> **Questions about Theoretical Analysis and Algorithm**
>
> Thank you for sharing your detailed and insightful questions and suggestions, they really help us to improve our paper.
>
> Q1: In the proof of Theorem 1 (Appendix A), why is Problem (15) equivalent to Problem (2). Providing an explicit solution for $t$ and $b$ is important for Theorems 1 and 2.
> * A1: Thanks for the question.
>
> Following the same definition $g(x;k;s)=(k^{\top}x^{1+s})^{\frac{1}{1+s}}$, in Problem (15), we have the simplified form ${\gamma}^{\top}x + \lambda\min({\gamma}^{\top}x)=g(x;{\gamma};0) + \lambda g(x;{\gamma};-\infty)$.
>
> In such a way, $b=\gamma^{\frac{1}{1+s}}$.
>
> However, the specific value of $t$ is an __implicit function and cannot be solved explicitly in closed form__. This is because according to the fact that the function $g$ is continuous with respect to $s$ over its entire domain and based on the intermediate value theorem for continuous functions, there must exist a $t$ such that the linear combination of the linear functions at the two endpoints equals.
>
> Nonetheless, we emphasize that the __subsequent methods and proof strategies are independent of the explicit solution for t__. As long as there exists a $t\neq0$, the Jansen gap exists, and as $\lambda$ increases, $t$ will also increase. We will include the discussion into the Theorem statements, meanwhile, we ensure our proof conclusion remain the valid.
>
> Q2: The organization of Lemma 2, Lemma 3, and Theorem 3 (Appendix D-F) is somewhat disorganized. Place Lemma 3 before Lemma 2 and rewrite the proof of Lemma 2 to explain why the conclusion can be derived.
>
> * A2: Thanks for the suggestions.
>
> For the first question, we will place Lemma 3 before Lemma 2.
>
> For the second question, we will re-write the beginning of the Lemma 2 as: after we get the conclusion from Lemma 3, we will have $r(\mu)<\infty$. Then we will proof for any $b\in M, c>0$, we have $r(\mu+cb)<\infty$. The rest of the parts remain the same.
>
> Q3: The authors should provide some intuitive explanations for the weight  sg  to better elucidate the experimental phenomena (Case study in Section 6.3). For instance, under what circumstances is  $s_g$  larger, and when is it smaller?
> * A3: Thanks for the suggestions.
>
> Intutively, $s_g=1-\mu_g$ is the negative showadow prices. In Lines 298-303, the high shadow price $\mu_g$ indicates that this constraint is the primary factor constraining accuracy optimization. Conversely, a low or zero shadow price suggests that the fairness constraint currently imposes little restriction on accuracy.
>
> We will revise the description to clarify the meaning of $s_g$​, indicating that a high $s_g$​ signifies that this constraint is the primary factor limiting fairness optimization for group $g$, whereas a low or zero $s_g$​ implies that the accuracy constraint for group $g$ currently has little impact on the overall optimization. Additionally, we will revisit this concept in the case study presented in Section 6.3.
>
> Q4: Can the authors provide a theoretical analysis of this bias? Alternatively, could the authors change the sampling-based ranking to random sampling, and test the impact of this bias on the convergence rate of Jensen gap?
> * A4: Intuitively, a larger Q provides a more accurate gradient estimation but also incurs higher computational costs. We have conducted experiments to evaluate the impact of Q and will present the results. The results were conducted under the same settings of analysis section.
>
>
>
> |Q| 50 | 100 | 200 | 300 | 400 | full (unbiased) |
> |--|--|--|--|--|--|--|
> | NDCG (%)| 1.08 | 1.08 | 1.15 | 1.19 | 1.19 | 1.29|
> | MMF (%)| 1.2 | 1.28 |2.18 |2.10 |2.29 |2.31|
>
> From the results, we observe that increasing the sample value Q leads to improvements in both accuracy and fairness performance. However, in LLM-based recommender systems, a larger Q significantly increases training time (with __each item requiring an additional 1.5 seconds__) and storage space.   Different applications should select appropriate Q values based on their specific accuracy, fairness requirements, and computational constraints.
> We will include these experimental results and discussion in the Appendix.

---

> > ### Author Response · Authors · 2024-11-21
> > **Questions about Experiments and Minor concerns**
> >
> > Q5: There are only two datasets utilized in the main results (Tables 1 and 2), which is insufficient. The authors might consider adding one or two widely-used datasets, such as Amazon-Electronic.
> >
> > * A5: Thanks for your suggestions.
> >
> > we __add a representative dataset Amazon-Electronic tested on all baselines and our method__ to the effectiveness of our methods and involve __new baseline__  Max-min sample[1], FOCF. For new dataset Amazon-Electronic, we test on the most advanced model BigRec, and the following is the results:
> > | Model| NDCG@5 | MRR@5 | MMF@5 | NDCG@10 | MRR@10 | MMF@10 | NDCG@20 | MRR@20 | MMF@20 |
> > |------------------|---------|-------|-------|---------|--------|--------|---------|--------|--------|
> > | UNI| 4.61| 4.3| 0.26| 4.93| 4.43| 0.25| 5.3 | 4.53| 0.21|
> > | DRO| 4.65| 4.34| 0.24| 4.96| 4.46| 0.24| 5.33| 4.57| 0.21|
> > | Prop | 4.63| 4.33| 0.26| 4.96| 4.47| 0.25| 5.33| 4.57| 0.21|
> > | SDRO | 4.6 | 4.29| 0.25| 4.92| 4.42| 0.24| 5.29| 4.52| 0.2|
> > | IFairLRS| 2.21| 2.06| 0.19| 2.46| 2.16| 0.17| 2.69| 2.22| 0.12|
> > | **Maxmin Sample (new)**| 4.6| 4.31| 0.27| 4.92| 4.44| 0.25| 5.31| 4.55| 0.21|
> > | **FairDual(Ours)** | __5.08__| __4.78__| __0.31__ | __5.43__| __4.92__| __0.3__ | __5.84__| __5.03__| __0.26__|
> >
> > From the results, we can see our methods can still outperform all the baselines, indicating the effectiveness of our methods.
> >
> > [1] J. D. Abernethy, P. Awasthi, M. Kleindessner, J. Morgenstern, C. Russell, and J. Zhang. Active sampling for min-max fairness in ICML 2022.
> >
> > Q6: Then, is it possible to apply popularity debias method to the proposed algorithm, e.g., Inverse Propensity Score (IPS)-based reweighting method?
> > * A6: It is a very insightful question. Due to the tight schedule, we conduct the experiments on a relatively light transformer-based SASRec [2] backbones and MIND datasets, here is the results on top5 results:
> >
> > For NDCG (accuracy):
> > |$\lambda$| 0.1 | 1 | 2 | 5 |
> > |--|------------------|---------|-------|-------|
> > |IPS|0.58|0.58|0.58|0.58|
> > |FairDual|0.53|0.6|0.57|0.67|
> > |FairDual+IPS|0.59|0.56|0.56|0.58|
> >
> > For MMF (fairness)
> > |$\lambda$| 0.1 | 1 | 2 | 5 |
> > |--|------------------|---------|-------|-------|
> > |IPS|12.63|12.63|12.63|12.63|
> > |FairDual|4.5|11.98|13.46|13.76|
> > |FairDual+IPS|10.9|12.4|12.4|14.36|
> >
> > From the results, we can observe that when the $\lambda$ is small, adding the IPS will increase the accuracy and fairness with a large margin due to the popularity bias. However, when $\lambda$ is large, the FairDual+IPS will not perform very good. This is because IPS will break the convergence condition of FairDual. Therefore, when $\lambda$ is large, it is preferable not to involve IPS. We will include the related experiments and discussion in the Appendix of the revised paper.
> >
> > Q6: Line 324 $c_{u,i}=−d(e_u,e_i)$, should there be $d(e_u,e_i)$?
> > * A6: Sorry for the confusion. In our definition, $d(x,y)=-x^{\top}y$ is the distance of x and y (negative similarities) and in recommendation, the more small distance will mean that x and y have more similarities, therefore $c_{u,i}=−d(e_u,e_i)=e_u^{\top}e_i$. We will consider to directly write $c_{u,i}=e_u^{\top}e_i$ by removing the $d()$ function.
> >
> > Q7: Line 325, the authors should suppose that $e_u$ and $e_i$ are normalized.
> > * A7: Thanks for the suggestion, we will modify it accordingly.
> >
> > Q8: Line 357, the L should be Q?
> > * A8: Yes! sorry for the typo, we will modify it accordingly.
> >
> > Q9: Line 979, the minus in  −I  should be placed at the loss term.
> > * A9: Thanks for the suggestion, we will modify it accordingly.
> >
> >
> > [2] Wang-Cheng Kang et al. "Self-Attentive Sequential Recommendation." in ICDM 2018.

---

> > > ### Comment · Reviewer_SoKV · 2024-11-21
> > > **Response to Author's Rebuttal**
> > >
> > > Thank you for your detailed response, which addressed most of my concerns:
> > >
> > > - I now fully understand the derivation in A1; this proof is clear and requires no modifications. I apologize for my earlier misunderstanding. From my perspective, the theoretical part of this paper should have no major issues.
> > > - I like the explanation of the shadow price in A3; this encapsulates the essence of using dual optimization to address this problem. I suggest that the authors highlight this explanation in their paper.
> > > - I thank the authors for the sampling-related experimental analysis provided in A4. From the results, it is clear that sampling introduces some gap, but addressing this issue exceeds the scope of this paper.
> > > - The authors' experimental results are very timely, and their findings on Amazon-Electronic align with my experience. The experiments involving IPS are also intriguing, as IPS is a standard technique for recommendation debiasing, and it is important to study the trade-off between debiasing and fairness. I thank the authors for their efforts and strongly recommend incorporating these insightful experiments into the discussion of the experimental results.
> > > - My final suggestion is that the authors should not label metrics such as NDCG, MRR, and MMF with percentage signs (%)! This is a very confusing practice that has led to misunderstandings among reviewers, including myself and Reviewer `w1tq`.
> > >
> > > Considering the above comments, I have increased my score.

---

> > > > ### Author Response · Authors · 2024-11-21
> > > >
> > > > Thanks for your hard work and final suggestions, we will not label metrics such as NDCG, MRR, and MMF with percentage signs (%) in the revised version. Your suggestion really helps us to improve our paper.
> > > >
> > > > Best,
> > > > Authors

---

### Official Review · Reviewer_REQg · 2024-11-04

**Soundness:** 2
**Presentation:** 2
**Contribution:** 3
**Rating:** 5
**Confidence:** 4

**Summary:**

This paper, through theoretical and empirical analysis, argues that existing methods with group max-min fairness (MMF) optimization for fairness-aware recommender systems will introduce Jensen gaps during model convergence when applying mini-batch sampling. It theoretically reformulates the MMF constraint objective as a group-weighted optimization objective and proposes a FairDual algorithm to minimize the Jensen gap. The effectiveness of FairDual is verified on two public datasets combined with three skeleton recommendation models.

**Strengths:**

S1: Sufficient support is provided for motivation through theoretical and empirical analysis.

S2: A new perspective based on group-weighted optimization is provided for MMF, and some corresponding theoretical insights are provided.

S3: Combining different skeleton models on multiple datasets provides various experimental results.

**Weaknesses:**

W1: The quality and clarity of the presentation need to be improved. Including some unclear statements that need to be revised, the organization of figures and tables needs to be revised, and some content coherence needs to be further revised. For example,
* For Formula 1, the meaning of the symbol $n_i$ seems missing. Secondly, based on the description in the second paragraph of Section 3, $L_k(u)$ is used to represent the recommendation list, and $K$ identifies the list length. However, in the description in the last paragraph, the statement "following the practice in time-aware RS, there may be users with interactions cu,i greater than the ranking size K, in which case we will only consider the least recent K interactions to represent their recent preferences" is confusing. Why is it emphasized that there are users with more than K interactions? Why must the most recent K interactions be kept instead of other numbers?

* Regarding the first paragraph of Section 4, the statement “In real-world scenarios, the number of users |U| is often large, and a mini-batch sampling strategy is often necessary due to the large computational costs. This involves dividing the |U| users into |U|/B batches and performing gradient descent methods on each batch” is also confusing. In my experience, in practice, grouping users into different batches for optimization is not usually adopted, i.e., each batch only contains a subset of users, and all interactions of each user are included. Secondly, the following description seems to use a random batch sampling. The purpose of emphasizing this aspect here is unclear.

* Regarding the second paragraph of Section 4.2 and Figure 1, as far as I understand, the convergence of the same group should be examined under different batch sizes to obtain the desired observations and conclusions.
* For the last paragraph of Section 5.2.1, the meaning of *P* is missing.

* Based on the current description of Section 5.2.2, I can't find any instructions on handling the first batch since it lacks a pre-batch to compute $g$. Secondly, does the operation of sampling $Q$ items bring noise or instability? This requires more discussion and experimental analysis.

* The current placement of figures and tables does not facilitate reading and needs to be placed as close to the corresponding description as possible.

* I like the first half of this paper. However, I am confused about why fairness must be associated with large recommendation models (especially large language recommendation models) after the methods section. On the one hand, this makes some of the treatments required for large language recommendation models appear abruptly. On the other hand, it is not conducive to evaluating the effectiveness of the proposed solution for fairness in a more general setting.

W2: The proposed FairDual lacks some deeper and more valuable insights. For example,
* Does it have the same performance or properties for other types of loss functions?

* Does it have the same behavior or properties as other fairness optimization constraints?

* How does it compare to existing work regarding storage space, computational complexity, and parameter size? Some static or dynamic group weighting methods discussed in related work seem lightweight. Is the additional overhead worthwhile?

* If it is not just about fairness at the item group level, does it apply to fairness at the user group level or even in situations where both users and items exist in groups?

W3: The current experimental setup and experimental results are not convincing enough.
* Representative datasets adopted by many previous fairness recommendation methods should be included more.

* Related to the previous concerns, the current version's selection criteria for baselines are confusing and not sufficiently representative. Skeleton models should not be constrained to be recommendation models related to large language models, and more research lines of fair recommendation methods mentioned in related work should be included as baselines, especially those aimed at designing group weighting.

* The current description of the implementation details is oversimplified, which is not conducive to reproducibility. Secondly, $\lambda$ is mentioned to range from 0 to 5, but in Figure 3 it is inclusive of 0 to 10.

**Questions:**

Please see the description in Weaknesses.

---

> ### Author Response · Authors · 2024-11-21
> **Summary of Rebuttal**
>
> Thank you for sharing your detailed and insightful questions and suggestions, they really help us to improve our paper.
>
> Firstly, we will summarize our changes in the rebuttal as follows:
>
> (1) We __recheck and review__ our quality and clarity of the __presentation__ according to your questions.
>
> (2) For the deeper and more valuable insights, we conduct the following discussion and rebuttal
>
> * We involve more discussion about our method is highly adaptable and can be generalized to various __group fairness form__.
> * We conduct the experiments on __other loss functions and fairness metrics__. The experiments also verify the effectiveness of our methods.
>
> (3) For the experimental setup and results:
>
> * We add three most widely used __non-LLMs-based recommendation backbones__ (BPR [1], GRU4Rec [2], SASRec [3]) on MIND dataset to show our methods are effective in a general setting.
>
>
> * We also __add a representative dataset Amazon-Electronic [4] tested on all baselines and our method__ to the effectiveness of our methods.
>
> *  We involve __four baselines__ covered from __optimizing-based__ methods,  and __group fair-aware__ recommendation methods. Note that group fair-aware recommendation models cannot well adapted into LLMs-based recommender models since they are often developed under pair-wise form of recommendation [1].
>
> __Optimizing-based__ baselines:
> [5] Max-min sample: applies optimizing techniques to dynamically sample groups.
> [6] FOCF: applies a fair-aware regularization loss of different groups into non-LLMs RS.
>
>  __Group fair-aware__ methods in recommendation:
> [7] Reg: Penalizes the squared diference between the average scores of two groups for all positive user-item pairs into non-LLMs RS.
> [8] FairNeg: A negative sampling way for pair-wise recommendation into non-LLMs RS.
>
> * We conduct more analysis about the __sample number $Q$, storage space, complexity, and parameter size__.
>
>
> Secondly, we want to emphasize that our paper primarily addresses the significant yet often overlooked bias (Jensen gap) that arises when optimizing fairness objectives in recommendation systems. We thoroughly analyze the reasons behind this bias and propose FairDual, a well-generalized and efficient algorithm to bridge the Jensen gap. Our approach is validated across three diverse datasets and six state-of-the-art backbones, compared against various types of baselines. We believe our paper can help other researchers explore its applicability to various loss functions, objectives, and fairness concepts.
>
>
> In summary, we kindly ask you to consider __both our theoretical contributions and real-industrial applications__ for the fairness communities. In the following responses, we will address your question __one by one__ to ensure clarity and thoroughness.
>
> [1] Steffen Rendle et al. "BPR: Bayesian Personalized Ranking from Implicit Feedback." in UAI 2009.
> [2] Yong Kiam Tan et al. "Improved Recurrent Neural Networks for Session-based Recommendations." in DLRS 2016.
> [3] Wang-Cheng Kang et al. "Self-Attentive Sequential Recommendation." in ICDM 2018.
> [4] R. He and J. McAuley. Ups and downs: Modeling the visual evolution of fashion trends with one-class collaborative filtering.
> [5] J. D. Abernethy, P. Awasthi, M. Kleindessner, J. Morgenstern, C. Russell, and J. Zhang. Active sampling for min-max fairness in ICML 2022.
> [6] Yao sirui et al. Beyond parity: Fairness objectives for collaborative filtering in Neurips 2017.
> [7] Toshihiro Kamishima and Shotaro Akaho. 2017. Considerations on recommendation independence for a find-good-items task.
> [8] Chen, Xiao et al. Fairly Adaptive Negative Sampling for Recommendations in WWW 2023.

---

> ### Author Response · Authors · 2024-11-21
> **Weakness regarding to the quality and clarity of the presentation**
>
> W1.1: For Formula 1, the meaning of the symbol $n_i$ seems missing. Why is it emphasized that there are users with more than K interactions? Why must the most recent K interactions be kept instead of other numbers?
>
> * A1.1: Apologies for the confusion.
>
> $n_i$ is defined in line 129, representing the number of groups to which item $i$ belongs. We will recall this definition when introducing Formula 1 for clarity.
>
> For the $K$ interactions, apologies for the confusion in notation. $K$ should be replaced with another parameter $H$, which represents the truncated user historical behavior numbers. We have conducted experiments (see Table 4 in the appendix) to demonstrate performance changes with respect to $H$. We will make the necessary corrections.
>
> W1.2: Dividing the |U| users into |U|/B batches and performing gradient descent methods on each batch” is also confusing. In my experience, each batch only contains a subset of users, and all interactions of each user are included. Secondly, the following description seems to use a random batch sampling, which is unclear.
>
> * A1.2: Sorry for the confusion.
>
> For the first question, we indeed use the settings you mentioned: each batch contains only a subset of users, and all interactions are utilized. Here, we simply wanted to emphasize that in each epoch, we apply random batch sampling with a batch size of B. To clarify, we will revise the sentence to: We apply the regular mini-batch sampling strategy with a batch size of B.
>
> Secondly, we emphasize this aspect because our algorithm does not depend on the order in which users arrive, unlike other debiasing online learning algorithms that may require specific user training sequences. We will compare our approach with existing methods and highlight that our algorithm can be seamlessly integrated into regular mini-batch strategies, offering broader applicability.
>
> W1.3: Regarding the second paragraph of Section 4.2 and Figure 1, as far as I understand, the convergence of the same group should be examined under different batch sizes to obtain the desired observations and conclusions.
>
> * A1.3: Thank you for your suggestions. Figure 1(a) was conducted with the same group size (G=7) under different batch sizes, while Figure 1(b) was conducted with the same batch size (B=32) under different group sizes. We will include these details in the experimental descriptions for better clarity.
>
> W1.4: For the last paragraph of Section 5.2.1, the meaning of  _P_  is missing.
> * A1.4: Apologies for the confusion. $P(w)$ refers to the predicted probability of the word $w$ generated by the LLMs. We will clarify this in the revised version.
>
> W1.5: Based on the current description of Section 5.2.2, I can't find any instructions on handling the first batch since it lacks a pre-batch to compute $g$. Secondly, does the operation of sampling  Q  items bring noise or instability? This requires more discussion and experimental analysis.
> * A1.5: Sorry for the confusion.
>
> For the first question, in Algorithm 1 line 7, we initialize $g$ as 0, which will not make an effect on first batch. We will illustrate it in Section 5.2.2.
>
> For the second question, intuitively, a larger Q provides a more accurate gradient estimation but also incurs higher computational costs. We have conducted experiments to evaluate the impact of Q and will present the results. The results were conducted under the same settings of analysis section.
>
>
> |Q| 50 | 100 | 200 | 300 | 400 | full (unbiased) |
> |--|--|--|--|--|--|--|
> | NDCG (%)| 1.08 | 1.08 | 1.15 | 1.19 | 1.19 | 1.29|
> | MMF (%)| 1.2 | 1.28 |2.18 |2.10 |2.29 |2.31|
>
>
> From the results, we observe that increasing the sample value Q leads to improvements in both accuracy and fairness performance. However, in LLM-based recommender systems, a larger Q significantly increases training time (with __each item requiring an additional 1.5 seconds__) and storage space.   Different applications should select appropriate Q values based on their specific accuracy, fairness requirements, and computational constraints. We will include these experimental results and discuss them in the appendix.
>
> W1.6: The current placement of figures and tables does not facilitate reading and needs to be placed as close to the corresponding description as possible.
> * A1.6: Thanks for your suggestions! We will put the figures and tables more close to the corresponding description as possible in the revised version.

---

> > ### Author Response · Authors · 2024-11-21
> > **The proposed FairDual lacks some deeper and more valuable insights**
> >
> > W2.1 Does it have the same performance or properties for other types of loss functions?
> >
> > * A2.1: Thanks for your question.
> >
> > In recommendation, another widely used loss function is the BPR loss [1], which aims to increase the distance between positive and negative samples. Interestingly, we find that our methods can also be applied to this loss, as BPR loss is a convex function with respect to positive items, and our dual formulation remains valid.
> >
> > We conduct the experiments in BPR model [1] backbones (Non-LLMs) on MIND dataset with the aforementioned new baselines to show our effectiveness. Note that FairNeg is only designed for the BPR loss, therefore, it is not tested on other backbones.
> >
> > |Models|NDCG@5|MRR@5|MMF@5|NDCG@10|MRR@10|MMF@10|NDCG@20|MRR@20|MMF@20|
> > |-------------------|--------|-------|--------|---------|--------|---------|---------|--------|---------|
> > |Prop|0.42|0.32|0.05|0.57|0.38|0.06|0.95|0.48|10.0|
> > |DRO|0.73|0.62|12.9|0.87|0.72|11.8|1.12|0.79|12.9|
> > |SDRO|0.67|0.61|3.88|0.84|0.68|6.87|1.04|0.73|12.03|
> > |IFairLRS|0.68|0.57|0.13|0.77|0.61|0.23|1.07|0.69|1.38|
> > |**FOCF(new)**|0.4|0.32|0.05|0.57|0.38|0.07|0.95|0.48|10.0|
> > |**Max-min sample(new)**|0.66|0.58|6.54|0.81|0.64|8.8|1.05|0.71|10.87|
> > |**Reg(new)**|0.67|0.61|3.27|0.83|0.67|5.89|1.06|0.73|11.25|
> > |**FairNeg(new)**|0.72|0.63|6.07|0.91|0.71|8.8|1.21|0.79|12.64|
> > |__FairDual(Ours)__|__0.76__|__0.64__|__11.84__|__0.94__|__0.72__|__13.87__|__1.27__|__0.81__|__14.6__|
> >
> > From the results, we observe that our method maintains strong performance even with BPR loss functions. We will include these experiments, alongside results from other backbone models, in the main body of the revised paper for comprehensive comparison.
> >
> > W2.2: Does it have the same behavior or properties as other fairness optimization constraints?
> > * A2.2: It is a good question.
> >
> > In Theorem 1, we demonstrate that our optimization objective is equivalent to the power-family fairness framework, which encompasses mainstream fairness definitions such as Entropy Fairness, $\alpha$-Fairness, and Theil Index [9]. Consequently, our method is __highly adaptable and can be generalized to various fairness objectives within this framework__.
> >
> >
> > We also test the performances of other fairness metric Gini Index Compared to the baselines (Table 1) on MIND datasets. Note smaller Gini Index means more fairness.
> > |Models|GINI@5|GINI@10|GINI@20|
> > |-------------------|--------|-------|--------|
> > |Prop|0.488|0.488|0.472|
> > |DRO|0.511|0.476|0.487|
> > |SDRO|0.503|0.478|0.453|
> > |IFairLRS|0.458|0.454|0.448|
> > |**FairDual(ours)**|**0.444**|**0.450**|**0.441**|
> >
> > From the results, we can observe that our model can still perform good on other fairness metrics.
> >
> > We believe our paper can help other researchers explore its applicability to various loss functions, and other fairness metrics, which is also our contributions to the communities.
> >
> >
> > W2.3: How does it compare to existing work regarding storage space, computational complexity, and parameter size? Some static or dynamic group weighting methods discussed in related work seem lightweight. Is the additional overhead worthwhile?
> >
> > * A2.3: Thanks for the question.
> >
> > Firstly, we all have parameters of the same magnitude (i.e., group size parameters (hundred level), which are in the range of hundreds and negligible compared to the backbone (million level)).
> > Our method only requires additional space for Q item embeddings and extra training time (Q * 1.5s). Applications can trade off Q based on available resources (as discussed in a previous response).
> >
> > Secondly, as mentioned in Table 3 of the original paper, although there is an additional time overhead per round, __our convergence speed accelerates by 30% compared to the best baseline__. This 30% improvement in convergence speed is highly significant for industrial applications, along with enhanced performance.
> >
> > W3.3：If it is not just about fairness at the item group level, does it apply to fairness at the user group level or even in situations where both users and items exist in groups?
> > * A3.3: Thank you for your question.
> >
> > Indeed, our method can be easily generalized to the user group level by replacing the adjacency matrix with a user-side equivalent while keeping the rest unchanged.
> >
> > For the two-sided form, it simply requires introducing two coefficients, $\lambda_1$ and $\lambda_2$, and applying two independent dual gradient descent updates as described in our algorithm. We will include a detailed discussion on this in the revised version.
> >
> >
> > [9] Lan, T., & Chiang, M. (2011). An axiomatic theory of fairness in resource allocation. _George Washington University, http://www. seas. gwu. edu/tlan/papers/fairness. pdf, Tech. Rep_.

---

> ### Author Response · Authors · 2024-11-21
> **The current experimental setup and experimental results are not convincing enough Part1**
>
> W3.1: The current description of the implementation details is oversimplified, which is not conducive to reproducibility. Secondly,  $\lambda$  is mentioned to range from 0 to 5, but in Figure 3 it is inclusive of 0 to 10.
> * A3.1: Sorry for the confusion.
>
> For the first question, we will add the following implementation details in the Appendix of revised version to help readers to reproduce our results:
>
> (1) For the environment, our experiments were implemented using Python 3.9 and PyTorch 2.0.1+cu117. All experiments were conducted on a server with an NVIDIA A5000 running Ubuntu 18.04.
>
> (2) The pre-processing steps are detailed in Appendix H. For the missing hyper-parameters,  we tune sample number $Q\in [50, 400]$ (results show in the previous responses), historical length $H\in [3,7]$ (results show in Table 4), freeze parameter updating gap $\beta\in[128, 3840]$.
>
> (3) To mitigate the impact of randomness, we set the temperature coefficient to 0.2 for the LLM and ran each model three times, taking the average of the results. Other LLMs settings are: the penalty for frequency is 0.0, and the penalty for presence is 0.0, the maximum generated token number to 1024.
>
> (4) For the Non-LLMs-RS backbones, we mainly reference the RecBole toolkit (https://github.com/RUCAIBox/RecBole). For the LLMs tuning, we reference the BigRec pipelines (https://github.com/SAI990323/BIGRec). __And we have also included our code in the supplementary materials to ensure reproducibility__.
>
> For the second question, we are sorry for the typo, the $\lambda$ should be tuned among [0,10]. We will revise it accordingly.
>
> W3.2 Representative datasets adopted by many previous fairness recommendation methods should be included more. Baselines are confusing and not sufficiently representative.
>
> * A3.2 Thanks for the question, as mentioned in summary, we __add a representative dataset Amazon-Electronic [4] tested on all baselines and our method__ to the effectiveness of our methods and involve __four baselines__ covered from __optimizing-based__ methods,  and __group fair-aware__ recommendation methods: : Max-min sample[5], FOCF [6], Reg[7] and FairNeg [8]. Note FOCF, Reg is designed for the non-LLMs RS models and FairNeg is designed for the pair-wise RS models.
>
> For new dataset Amazon-Electronic, we test on the most advanced model BigRec, and the following is the results:
> | Model| NDCG@5 | MRR@5 | MMF@5 | NDCG@10 | MRR@10 | MMF@10 | NDCG@20 | MRR@20 | MMF@20 |
> |------------------|---------|-------|-------|---------|--------|--------|---------|--------|--------|
> | UNI| 4.61| 4.3| 0.26| 4.93| 4.43| 0.25| 5.3 | 4.53| 0.21|
> | DRO| 4.65| 4.34| 0.24| 4.96| 4.46| 0.24| 5.33| 4.57| 0.21|
> | Prop | 4.63| 4.33| 0.26| 4.96| 4.47| 0.25| 5.33| 4.57| 0.21|
> | SDRO | 4.6 | 4.29| 0.25| 4.92| 4.42| 0.24| 5.29| 4.52| 0.2|
> | IFairLRS| 2.21| 2.06| 0.19| 2.46| 2.16| 0.17| 2.69| 2.22| 0.12|
> | **Maxmin Sample (new)**| 4.6| 4.31| 0.27| 4.92| 4.44| 0.25| 5.31| 4.55| 0.21|
> | **FairDual(Ours)** | __5.08__| __4.78__| __0.31__ | __5.43__| __4.92__| __0.3__ | __5.84__| __5.03__| __0.26__|
>
> For other two dataset, the following is the new baseline results:
>
> For MIND dataset
> | Model| NDCG@5 | MRR@5 | MMF@5 | NDCG@10 | MRR@10 | MMF@10 | NDCG@20 | MRR@20 | MMF@20 |
> |------------------|---------|-------|-------|---------|--------|--------|---------|--------|--------|
> | **Maxmin Sample (new)**| 0.98| 0.75| 2.25| 1.49| 0.96| 1.71| 2.19 | 1.15| 3.13|
> | **FairDual(Ours)**| __1.15__| __0.88__| __2.82__| __1.69__| __1.11__| __2.99__| __2.28__| __1.27__| __3.39__|
>
> For Amazon-book dataset
> | Model| NDCG@5 | MRR@5 | MMF@5 | NDCG@10 | MRR@10 | MMF@10 | NDCG@20 | MRR@20 | MMF@20 |
> |------------------|---------|-------|-------|---------|--------|--------|---------|--------|--------|
> | **Maxmin Sample (new)**| 2.49| 2.31| 6.80| 2.72| 2.43| 6.80| 2.97 | 2.74| 7.5|
> | **FairDual(Ours)**| __3.11__| __2.88__| __8.90__| __3.31__| __2.96__| __9.00__| __3.60__| __3.04__| __8.89__|
>
> From the results, we can also see that our model can outperform all the baselines, showing our effectiveness. We will add the results in the main body of the revised paper.
>
> __For other baselines FOCF, Reg and FairNeg, we compare them in the following non-LLM backbone model experiments.__

---

> ### Author Response · Authors · 2024-11-21
> **Experiment Part2**
>
> W3.3 Recommendation models related to large language models, and more research lines of fair recommendation methods mentioned in related work should be included as baselines.
> * A3.3: Thank you for your suggestions. You are correct that our backbone models do not necessarily need to be LLMs; we use LLMs in this case because the small batch size poses greater challenges for LLM-based recommender models in addressing the Jansen gaps.
>
> As mentioned in previous responses, we add three most widely used __non-LLMs-based recommendation backbones__ (BPR [1], GRU4Rec [2], SASRec [3]) on MIND dataset to show our methods are effective in a general setting.  We already report the results on BPR in previous responses.
>
> The results of GRU4Rec:
> | Models| NDCG@5 | MRR@5 | MMF@5 | NDCG@10 | MRR@10 | MMF@10 | NDCG@20 | MRR@20 | MMF@20 |
> |-------------------|--------|-------|-------|---------|--------|--------|---------|--------|--------|
> | UNI| 0.39| 0.36| 5.08| 0.55| 0.42| 6.44| 0.83| 0.5| 9.08|
> | Prop| 0.42| 0.35| 7.94| 0.63| 0.44| 10.19| 0.9| 0.51| 13.1|
> | DRO| 0.56| 0.56| 0.86| 0.76| 0.64| 5.56| 1.13| 0.71| 10.7|
> | SDRO| 0.45| 0.36| 11.42 | 0.67| 0.44| 12.05| 0.97| 0.53| 13.15|
> | IFairLRS| 0.45| 0.38 | 7.12  | 0.68| 0.47 | 9.21 | 1.02| 0.56 | 11.70 |
> | **FOCF (new)**| 0.56| 0.41| 5.62| 0.79| 0.63| 7.11| 1.1| 0.7| 10.29|
> | **Maxmin sample(new)**| 0.43| 0.33| 10.9| 0.62| 0.41| 14.27| 0.91| 0.48| 13.06|
> | **Reg(new)**| 0.45| 0.37| 6.93| 0.67| 0.46| 8.6| 1.02| 0.55| 10.92|
> | **FairDual (Ours)**| __0.59__| __0.47__| __12.13__ | __0.85__| __0.68__| __12.77__| __1.16__| __0.76__| __14.09__|
>
> For the results of SASRec:
> | Models| NDCG@5 | MRR@5 | MMF@5 | NDCG@10 | MRR@10 | MMF@10 | NDCG@20 | MRR@20 | MMF@20 |
> |------------------|--------|-------|-------|---------|--------|--------|---------|--------|--------|
> | UNI| 0.59| 0.5| 10.43 | 0.76| 0.56| 11.91| 1.09| 0.65| 12.94|
> | Prop| 0.54| 0.45| 11.69 | 0.8| 0.55| 12.1| 1.16| 0.57| 13.01|
> | DRO| 0.54| 0.4| 8.07| 0.72| 0.47| 11.34| 1.11| 0.57| 12.26|
> | SDRO| 0.49| 0.4| 10.66 | 0.74| 0.49| 11.64| 1.09| 0.59| 14.02|
> | IFairLRS| 0.58| 0.57| __12.63__ | 0.60| 0.58| 12.35| 0.62| 0.58| 13.73|
> | **FOCF(new)**| 0.47| 0.46| 10.52 | 0.5| 0.47| 12.73| 0.53| 0.48| 14.46|
> | **Minmax_SGD(new)**| 0.56| 0.47| 9.05| 0.74| 0.54| 12.45| 1.09| 0.64| 14.06|
> | **Reg(new)**| 0.47| 0.38| 9.42| 0.7| 0.47| 9.52| 1.03| 0.55| 10.91|
> | **FairDual (Ours)**| __0.64__| __0.63__| 11.98 | __0.78__| __0.64__| __13.08__| __1.31__| __0.67__| __14.51__|
>
>
> From the results, we can also see that our model can outperform all the baselines except MMF@5 in SASRec, showing our effectiveness on non-LLMs RS models. We will add the results in the main body of the revised paper.
>
> Thanks again for your question and suggestions.

---

> ### Author Response · Authors · 2024-11-24
> **Deadline is approching**
>
> Dear reviewer REQg,
>
> As the discussion deadline will end in less than three days, we would like to know whether our responses have adequately addressed your concerns. Please do not hesitate to reach out if you have any further questions or need clarification before the deadline. We greatly appreciate your dedicated time and effort in reviewing our work.
>
> Best regards,
>
> Authors

---

> > ### Author Response · Authors · 2024-11-27
> >
> > Dear Reviewer REQg,
> >
> > We have revised the PDF and would like to know if our experimental results and discussions address your concerns. Since the rebuttal period has been extended, please feel free to reach out if you have any additional questions or concerns to discuss.
> >
> > Best regards,
> >
> > Authors

---

> ### Comment · Reviewer_REQg · 2024-12-02
>
> I am grateful to the authors for providing a detailed response that addresses my concerns.
>
> If the evaluation is based on the revised version, my three concerns are:
> * Re-polishing the entire paper was necessary because there were numerous revised sections. For example, the description in the last paragraph of the first section was not updated and was inconsistent with the abstract.
>
> * To avoid the abruptness of the content and too many necessary experimental results being placed outside the main text, I suggest that the authors focus their perspective and writing on a specific type of recommendation scenario, such as group fairness issues around recommendations based on large language models or sequence recommendations.
>
> * To verify the authors’ description of the advantages of FairDual on an industrial scale, it may be necessary to conduct experiments on some corresponding datasets, such as some datasets provided by the industry commonly used in click-through rate prediction tasks.
>
> Other minor issues: 1) Results from previous lightweight work on group fairness should be included in the efficiency experiments; 2) Is the added BPR baseline based on the original form of matrix factorization? This may be ambiguous since it is a loss function compatible with many models. 3) some recommendation models based on large language models are compatible with pairwise losses.
>
> If the evaluation is based on the original submission status, the revisions that need to be included may be far beyond what is acceptable for a normal camera-ready version and numerous experimental results may lack a second review.
>
> Considering the above comments, I will increase my score appropriately, but still do not think this submission is at an unquestionably acceptable level in this venue.

---

> ### Author Response · Authors · 2024-12-02
>
> Dear reviewer,
>
> Thanks a lot for your hard work and your suggestions. It really helps us improve our paper.
>
> Thank you for your suggestion regarding re-polishing the paper and placing additional experiments. In the revised version, we will refine the entire paper, include the efficiency experiments in the main text, and move the non-LLM experiments to the Appendix.
>
> For the second concern,
> (1) Our three datasets, MIND and Amazon, are industrial-scale datasets that can also be utilized for click-through rate prediction tasks [1]. Additionally, our efficiency experiments demonstrate that our methods are suitable for industrial-scale applications.
>
> (2) Typically, recommendation systems involve several key tasks, such as rating prediction, click-through rate prediction, and ranking tasks [2]. In this paper, we primarily **focus on fair ranking tasks (Section 3 Formulation), which are crucial and among the most widely used applications [3]**. In the revised version, we place __greater emphasis on the ranking task settings__. Moreover, we want to emphasize that our methods can also be easily extended to other tasks, as mentioned in our previous responses, which we believe will provide significant inspiration to the research community.
>
> For the minor concerns:
>
> Q1: Is the added BPR baseline based on the original form of matrix factorization?
> A1: Yes, the original BPR model is based on matrix factorization, we will emphasize it in the revised version.
>
> Q2: some recommendation models based on large language models are compatible with pairwise losses.
> A2: Due to the substantial computational costs of LLMs, recent LLM-based RS are often trained on small subsets of data, whereas traditional models can be trained on full datasets, often achieving comparable performance to LLMs on certain datasets, as noted in [4].
>
> In summary, we want to emphasize that, **in accordance with ICLR policy, the evaluation should be based on the revised version**, and we assure you that your new concerns regarding content organization and clarifications will be appropriately addressed in the camera-ready version (if accepted). __We kindly ask you to consider both our contributions for identifying and mitigating the ignoring Jensen gap for the fairness and RS communities__. We believe our paper can help other researchers explore its applicability to various applications.
>
> Best regards,
>
> Authors
>
>
> [1] Guorui Zhou, Xiaoqiang Zhu, Chenru Song, Ying Fan, Han Zhu, Xiao Ma, Yanghui Yan, Junqi Jin, Han Li, and Kun Gai. 2018. Deep Interest Network for Click-Through Rate Prediction. In Proceedings of the 24th ACM SIGKDD International Conference on Knowledge Discovery; Data Mining (KDD '18). Association for Computing Machinery, New York, NY, USA, 1059–1068.
> [2] Ko, H., Lee, S., Park, Y., & Choi, A. (2022). A survey of recommendation systems: recommendation models, techniques, and application fields. _Electronics_, _11_(1), 141.
> [3] Li, Y., Chen, H., Xu, S., Ge, Y., Tan, J., Liu, S., & Zhang, Y. (2022). Fairness in recommendation: A survey. _arXiv preprint arXiv:2205.13619_.
> [4] K. Bao, J. Zhang, W. Wang, Y. Zhang, Z. Yang, Y. Luo, F. Feng, X. He, and Q. Tian. A bi-step grounding paradigm for large language models in recommendation systems. arXiv preprint arXiv:2308.08434, 2023a.

---

### Author Response · Authors · 2024-11-22
**Revised pdf and summary**

Dear reviewers:

   Thanks for your hard work, your suggestions really help us to improve our paper. We revised our paper according to your suggestions (**revised parts are marked as blue**) and **re-upload our modified pdf**.

We will summarize our changes as follows:

(1) For the main body:
* We __recheck and review__ our quality and clarity of the __presentation__ according to your questions.
* We have incorporated experimental results on __new datasets, additional baselines (including different loss functions), and updated backbones__, as outlined in our rebuttal.
* We add more intuitive descriptions to explain our weight.

(2) For the Appendix:
* We re-polish our proof organization to help readers to understand.
* We involve more discussion about how our method is highly adaptable and can be generalized to various __group fairness forms, and different fairness constraints__.
* We add more detailed implementation details, baseline, and backbone details to enhance reproducibility.
* We conduct the analysis and experiments on the effect of **sample size Q, computational costs, other fairness metric,s and popularity bias**.

Finally, we want to emphasize that our paper primarily addresses the significant yet often overlooked bias (Jensen gap) that arises when optimizing fairness objectives in recommendation systems. We thoroughly analyze the reasons behind this bias and propose FairDual, a well-generalized and efficient algorithm to bridge the Jensen gap. We kindly ask you to consider __both our theoretical contributions and real-industrial applications__ for the fairness communities.

If you have any questions, please be free to ask them before the deadline (Nov. 26), we will answer them as soon as possible.

Best,

Authors

---

### Author Response · Authors · 2024-11-25
**Rebuttal deadline is approching**

Dear Reviewers,

Thanks for your hard work reviewing our paper and for your suggestions. As the discussion deadline will end in less than two days, however, only two reviewers responded to our rebuttal.

We would like to know whether our responses have adequately addressed your concerns. Please do not hesitate to reach out if you have any further questions or need clarification before the deadline. We greatly appreciate your dedicated time and effort in reviewing our work.

Best regards,

Authors

---

### Meta-Review · Area_Chair_WcE4 · 2024-12-24

**Metareview:**

This paper studies the group max-min fariness (MMF) constrained optimisation probem in recommendation. The authors theoretically show the MMF-constrained objective can be reformulated as a group-weighted optimization objective. Then, they propose a dual optimization method, named FairDual,to minimise the Jensen gap. Some theoretical analysis of the proposed method have been performed, and extensive experiments have been performed on public datasets to demonstrate the effectiveness of the proposed method.

Overall, this paper is well motivated, and the proposed method is novel. The proposed method is solid with a guaranteed bound for Jesen gap, and the experiment also showed that the proposed method indeed has lower gap than other baselines. Moreover, the authors also conduct comprehensive evaluations, including the effectiveness, Jensen gap analysis, case study, training efficiency, and parameter analysis.

**Additional Comments On Reviewer Discussion:**

In the rebuttal, the authors modify the presentation of the paper according to the review comments, provide more experimental results on new datasets, additional baselines, and updated backbones. Moreover, they also perform analysis on the sample size, computational cost, fairness metric, and popularity bias. The authors have well addressed the reviewer's concerns regarding with the theoretical analysis, sampling-related experimental analysis, the assumption of convexity, the value of NDCG, the baselines, and the statistical significance of MRR.

---

### Decision · Program_Chairs · 2025-01-22

Accept (Poster)